# Diabetes Mellitus in Non-Functioning Adrenal Incidentalomas: Analysis of the Mild Autonomous Cortisol Secretion (MACS) Impact on Glucose Profile

**DOI:** 10.3390/biomedicines12071606

**Published:** 2024-07-18

**Authors:** Alexandra-Ioana Trandafir, Adina Ghemigian, Mihai-Lucian Ciobica, Claudiu Nistor, Maria-Magdalena Gurzun, Tiberiu Vasile Ioan Nistor, Eugenia Petrova, Mara Carsote

**Affiliations:** 1PhD Doctoral School of “Carol Davila” University of Medicine and Pharmacy, 020021 Bucharest, Romania; alexandra-ioana.trandafir@drd.umfcd.ro; 2Department of Clinical Endocrinology V, “C.I. Parhon” National Institute of Endocrinology, 011863 Bucharest, Romania; adina.ghemigian@umfcd.ro (A.G.); eugenia.petrova@umfcd.ro (E.P.); carsote_m@hotmail.com (M.C.); 3Department of Endocrinology, “Carol Davila” University of Medicine and Pharmacy, 020021 Bucharest, Romania; 4Department of Internal Medicine and Gastroenterology, “Carol Davila” University of Medicine and Pharmacy, 020021 Bucharest, Romania; 5Department of Internal Medicine I and Rheumatology, “Dr. Carol Davila” Central Military University Emergency Hospital, 010825 Bucharest, Romania; 6Department 4-Cardio-Thoracic Pathology, Thoracic Surgery II Discipline, “Carol Davila” University of Medicine and Pharmacy, 050474 Bucharest, Romania; 7Thoracic Surgery Department, “Dr. Carol Davila” Central Military University Emergency Hospital, 010242 Bucharest, Romania; 8Cardiology Discipline, “Carol Davila” University of Medicine and Pharmacy, 020021 Bucharest, Romania; magdalena.gurzun@umfcd.ro; 9Laboratory of Non-Invasive Cardiovascular Exploration, “Dr. Carol Davila” Central Military University Emergency Hospital, 010242 Bucharest, Romania; 10Medical Biochemistry Discipline, “Iuliu Hatieganu” University of Medicine and Pharmacy, 400347 Cluj-Napoca, Romania; tiberiu.nistor@umfcluj.ro

**Keywords:** adrenal, tumour, incidentaloma, glucose, endocrine, diabetes, metabolic syndrome, adrenalectomy

## Abstract

Non-functioning adrenal incidentalomas (NFAIs) have been placed in relationship with a higher risk of glucose profile anomalies, while the full-blown typical picture of Cushing’s syndrome (CS) and associated secondary (glucocorticoid-induced) diabetes mellitus is not explicitly confirmed in this instance. Our objective was to highlight the most recent data concerning the glucose profile, particularly, type 2 diabetes mellitus (T2DM) in NFAIs with/without mild autonomous cortisol secretion (MACS). This was a comprehensive review of the literature; the search was conducted according to various combinations of key terms. We included English-published, original studies across a 5-year window of publication time (from January 2020 until 1 April 2024) on PubMed. We excluded case reports, reviews, studies on T1DM or secondary diabetes, and experimental data. We identified 37 studies of various designs (14 retrospective studies as well 13 cross-sectional, 4 cohorts, 3 prospective, and 2 case–control studies) that analysed 17,391 individuals, with a female-to-male ratio of 1.47 (aged between 14 and 96 years). T2DM prevalence in MACS (affecting 10 to 30% of NFAIs) ranged from 12% to 44%. The highest T2DM prevalence in NFAI was 45.2% in one study. MACS versus (non-MACS) NFAIs (n = 16) showed an increased risk of T2DM and even of prediabetes or higher fasting plasma glucose or HbA1c (no unanimous results). T2DM prevalence was analysed in NFAI (N = 1243, female-to-male ratio of 1.11, mean age of 60.42) versus (non-tumour) controls (N = 1548, female-to-male ratio of 0.91, average age of 60.22) amid four studies, and two of them were confirmatory with respect to a higher rate in NFAIs. Four studies included a sub-group of CS compared to NFAI/MACS, and two of them did not confirm an increased rate of glucose profile anomalies in CS versus NFAIs/ACS. The longest period of follow-up with concern to the glycaemic profile was 10.5 years, and one cohort showed a significant increase in the T2DM rate at 17.9% compared to the baseline value of 0.03%. Additionally, inconsistent data from six studies enrolling 1039 individuals that underwent adrenalectomy (N = 674) and conservative management (N = 365) pinpointed the impact of the surgery in NFAIs. The regulation of the glucose metabolism after adrenalectomy versus baseline versus conservative management (n = 3) was improved. To our knowledge, this comprehensive review included one of the largest recent analyses in the field of glucose profile amid the confirmation of MACS/NFAI. In light of the rising incidence of NFAI/AIs due to easier access to imagery scans and endocrine evaluation across the spectrum of modern medicine, it is critical to assess if these patients have an increased frequency of cardio-metabolic disorders that worsen their overall comorbidity and mortality profile, including via the confirmation of T2DM.

## 1. Introduction

Non-functioning adrenal incidentalomas (NFAIs) have been placed in relationship with a higher risk of glucose profile anomalies, while the full-blown typical picture of Cushing’s syndrome (CS) and associated secondary (glucocorticoid-induced) diabetes mellitus is not explicitly confirmed in this instance. Hence, patients diagnosed with NFAIs should be assessed with regard to the cardio-metabolic features, including type 2 diabetes mellitus (T2DM). Diabetes, a widespread, life-threatening medical condition that is predicted to increase dramatically in the future, contributes to the fast-worsening global health crisis [1]. According to estimates from the International Diabetes Federation (IDF), there are currently 537 million adults worldwide diagnosed with the condition, and this number will rise from 643 million in 2030 to 783 million in 2045 [2]. T2DM represents approximately 90–95% of all cases of the disease [3].

T2DM is a complex metabolic condition marked by long-term hyperglycaemia (if uncontrolled), insulin resistance, and pancreatic β-cell dysfunction. Hyperglycaemia-induced oxidative stress is an important cause of diabetic complications that are commonly categorized into two main groups: microvascular disorders, including retinopathy, nephropathy, and neuropathy, and macrovascular ailments (cardiovascular disease) [4,5]. People with diabetes have a 2- to 4-fold increase in mortality [6,7,8]. Myocardial infarction is the main cause of cardiovascular death in DM, while acute myocardial infarction accounts for over one-third of all deaths worldwide [9,10].

Additionally, the glucose-regulating metabolic pathways are widely recognized to be impacted by both chronic exposure to exogenous glucocorticoids and overt hypercortisolism. Thus, adrenal gland hyper-function causing hypercortisolism induces a dysregulation of the glycaemic metabolism. The overproduction of cortisol promotes the expression of multiple important gluconeogenesis enzymes, which subsequently increase liver glucose synthesis. Moreover, by directly interfering with the insulin receptor signalling pathways and indirectly by promoting lipolysis and proteolysis, it reduces insulin sensitivity, as a contributor to T2DM development [11,12]. For instance, CS patients may have different degrees of abnormal glucose metabolism, such as impaired glucose tolerance (IGT), fasting plasma glucose (FPG), and diabetes. About 20–45% of CS patients are diagnosed with diabetes, and nearly 70% are thought to have IGT [13]. This comes in addition to the other complex multidisciplinary clinical elements of CS, such as cardio-metabolic traits including high blood pressure, dyslipidaemia as well as osteoporosis and increased fracture risk, digestive complications, etc., that should be differentiated from non-CS causes [14,15,16,17].

However, even mild hypercortisolemia (previously known as subclinical hypercortisolism, subclinical CS or hidden hypercortisolism) is associated with alterations in the glucose metabolism [18,19,20]. This condition is the most common hormonal anomaly in individuals with NFAI, affecting 10% up to 30% of the subjects confirmed with NFAIs [21,22,23,24]. Of note, NFAIs may be registered under the general term of “adrenal incidentaloma” (AI), which remains less accurate for daily endocrine practice nowadays. The prevalence of NFAIs varies with the patients’ age and detection method, with less than 1% of NFAIs being detected in younger persons, and up to 10% are found in those over the age of 70 [25,26]. While unilateral incidentalomas are generally more common, 10 to 15% of patients may have bilateral adrenal masses [27,28]. Overall, 80% of the adrenal masses are benign [29].

In 2023, The European Society of Endocrinology (ESE) published the latest clinical recommendations that re-defined the term of “mild autonomous cortisol secretion” (MACS) for revealing the mentioned sub-group amid NFAIs that display biochemical, mild, persistent hypercortisolism and abnormal dexamethasone suppression test (DST) results (plasma morning cortisol after DST > 1.8 µg/dL) but without the traditional picture of CS [30]. However, the adequate tests and associated cut-offs to diagnose MACS are still an open issue. The 1 mg DST represents the standard for dynamic testing to determine MACS; the cut-offs for the second-day plasma cortisol are between 1.8 and 5 µg/dL [31,32]. DST accuracy has been compared to the urine steroid metabolomics in identifying MACS in NFAI/AI patients, a method that is not feasible in any endocrine centre of adrenal assessments [33,34].

Growing evidence indicates that there is a correlation between the patients displaying a MACS profile (amid the large panel of NFAIs) and a more severe spectrum of metabolic characteristics, including T2DM, versus typical NFAIs (non-MACS); but this spectrum remains heterogeneous, and many data are still under debate [35,36,37,38,39,40]. Despite conflicting results, MACS patients have been reported to have an increased risk of mortality, hence the importance of specifically addressing this topic [41,42]. MACS management is still a contentious issue; convincing data showed that MACS subjects who underwent adrenalectomy showed a significant improvement in T2DM and hypertension when compared to those who received conservative care [43,44,45].

Our objective was to highlight the most recent data concerning the glucose profile, particularly, T2DM, in NFAIs with/without MACS.

This was a comprehensive review of the literature; the research was conducted according to various combinations of key terms, as shown below. We included English-published, original studies across a 5-year window of publication time (from January 2020 until 1 April 2024) on PubMed. We excluded case reports, case series, experimental data, reviews, non-human studies, editorials, letters to the editor, reviews, meta-analyses, and conference abstracts as well as other non-full-length articles, studies on type 1 diabetes mellitus and other secondary forms of diabetes, cohorts on CS that did not introduce specific data on NFAI/AIs sub-groups and studies enrolling patients diagnosed with Cushing’s disease, Conn syndrome, pheochromocytoma, and adrenocortical cancer (Figure 1).

## 2. Results

We identified 36 original papers evaluating the glucose metabolism in patients with NFAI and/or MCAS. These studies (retrospective studies = 14, cross-sectional studies = 13, cohort studies = 4, prospective studies = 3, and case-control studies = 2) included a total of 17,391 individuals, with a higher prevalence of females (females = 10,363; males = 7028), aged between 14 and 96 years. This study-focused analysis (n = 37, N > 17,000) showed that patients with AI/NFAIs have an increased risk for T2DM versus MACS, respectively, versus controls. The prevalence of T2DM among patients with MACS ranged from 12% to 44%. Moreover, the highest prevalence of T2DM among subjects with NFAI was found up to 45.2%. The longest period of follow-up with concern to the glycaemic profile in AIs was 10.5 years, and one cohort showed a significant increase in the rate of T2DM at 17.9% compared to the baseline value of 0.03%. Additionally, inconsistent data from six studies enrolling 1039 individuals that underwent adrenalectomy (N = 674) and conservative management (N = 365) pinpointed an impact of adrenalectomy in NFAIs that is not very clear yet. In three of these studies, an improvement in the regulation of glucose metabolism after surgery was confirmed versus baseline versus conservative management. Of note, all these mentioned studies used the American Diabetes Association (ADA) criteria for the diagnosis of diabetes (across different years) [46,47,48,49,50,51].

### 2.1. Τ2 DM Prevalence in Patients Diagnosed with MACS versus (Non-MACS) NFAIs

Sixteen studies evaluated the glucose profile in patients diagnosed with MACS versus those with (non-MACS) NFAI (mean age was of 63.72 versus 61.83 years), and three of them had a control (non-NFAI) group. T2DM prevalence in MACS varied from 12% to 44%. However, the findings were not unanimous; most of the studies identified a statically significant difference in the prevalence of T2DM between MASC and NFAI, while three studies identified differences with concern to FPG and to glycated haemoglobin (HbA1c) in another study (Table 1).

All the mentioned studies used serum cortisol after 1 mg DST (c-1 mg-DST) ≥ 1.8 µg/dL or >1.9 µg/dL to diagnose MACS, except one study that used c-1 mg-DST > 5 µg/dL or a range between 1.8 and 5 μg/dL plus at least one of the following: low ACTH (Adrenocorticotropic Hormone), increased 24 h urinary free cortisol (UFC), absence of the diurnal cortisol rhythm, and plasma morning cortisol level after 2 days of low-dose (2 mg/day) dexamethasone suppression (LDDST) of >1.8 μg/dL. Various ADA criteria [FPG > 126 mg/dL, 2 h-plasma glucose ≥ 200 mg/dL during oral glucose tolerance test (OGTT), HbA1c ≥ 6.5%, a classic hyperglycaemic crisis, random plasma glucose ≥ 200 mg/dL or prior diagnosis of diabetes and associated therapy] were used to diagnose T2DM. Of note, the designation of “MACS” was not unanimously used; alternatively, autonomous cortisol secretion (ACS) included subclinical CS and even CS [52,53,54,55,56,57,58,59,60,61,62,63,64,65,66,67].

For instance, a total of 194 individuals with various adrenal tumours were included in one retrospective study: 97 patients with ACS and 97 patients with NFAI. ACS was further divided into subclinical CS (N = 80) and overt CS (N = 17). The prevalence of T2DM was statistically significantly higher in patients with ACS than NFAI (44% versus 22%, *p* = 0.002). As a potential bias, the dual inclusion of both subclinical and clinical CS in the same ACS category might influence the diabetes prevalence [58]. Similarly, another retrospective cohort confirmed that T2DM was more common in MACS (N = 56 patients, representing 18.9%) than in NFAI (N = 239 subjects, representing 80.1%) in terms 41% versus 23% (*p* < 0.01). Moreover, MACS subjects more frequently underwent one or two combined antidiabetic therapy(s) than NFAI individuals (*p* < 0.01). No correlation between c-1 mg-DST and FPG in patients with MACS and NFAI (*p* > 0.05) was established. However, there was a positive correlation between UFC and FPG in the MACS sub-group [56]. These results were also supported by a multicentre retrospective analysis that included patients with NFAI (N = 305) and ACS (N = 337). The frequency of T2DM was higher in ACS than NFAI (32.1% versus 24.3%, *p* = 0.031) [60]. Other data highlighted a prevalence of 16.7% (N = 83 patients with ACS) versus 8.5% (N = 82 subjects with NFAIs) [66], respectively, of 28.3% versus 16% (*p* < 0.001) according to another cohort [63].

On the contrary, we mention some non-confirmatory studies. For example, a retrospective analysis identified 478 patients with NFAI and 231 patients with ACS. T2DM was diagnosed in 24.3% (N = 172) of the subjects with any AI. Individuals with T2DM had higher levels of midnight salivary cortisol (*p* = 0.010) and UFC (*p* = 0.039) than subjects without T2DM. Between patients with ACS and NFAI, there was no statistically significant difference in T2DM prevalence (27.7% versus 22.6%, *p* = 0.137), with an odds ratio (OR) = 1.31 (95% CI: 0.92–1.88). Nevertheless, glycaemic control was not as good in ACS patients as in NFAI patients. HbA1c levels and FPG values were statistically significantly higher in ACS than NFAI patients (6.5 ± 1.4 versus 6.1 ± 0.9%, *p* = 0.005, respectively, 112 ± 35.6 versus 105 ± 29 mg/dL, *p* = 0.004) [54]. A smaller study sample size (N = 30 patients with ACS versus 45 subjects with NFAI) showed a similar prevalence of T2DM between the two groups (40% versus 31.1%, *p* = 0.43) [67], and another cohort conducted by Falcetta et al. [62] found similar data of T2DM prevalence between ACS and NFAIs (17.3% in ACS versus 18.7% in NFAI, *p* = 0.786) [62], as well as Singh’s [61] cohort (N = 443, 41.9% versus 40.1%, *p* = 0.801) [61].

Regarding FPG, lipid profile, body mass index (BMI), and the diagnosis of T2DM, there was no statistically significance difference between the NFAI and ACS sub-groups (*p* > 0.05). Regression analysis revealed no statistically significant correlation between the presence of T2DM and BMI when stratified by BMI in ACS patients (OR = 1; 95% CI: 0.90–1.12, *p* = 0.969). The diagnosis of T2DM (involving 12% of the patients) did not correlate with the size of the AI, nor did it correlate with the NFAI or ACS group (*p* > 0.05) [64]. Moreover, in one study, the predictive power of DST for comorbidities associated with autonomous cortisol release in AI was assessed as follows. To define cortisol suppression, three different DST thresholds (of 1.8, 3, and 5 µg/dL) were evaluated. A total of 823 patients with AI were included in the study. Of these, 83.7% (N = 650) had one or more potential ACS-related comorbidities; T2DM was one of these ailments, with a frequency of 26%. T2DM was more prevalent in those with c-1 mg-DST ≥ 1.8 µg/dL (OR = 1.6, 95% CI: 1.2–2.2); in those with cortisol values ≥ 3.0 µg/dL (OR = 1.7, 95% CI: 1.1–2.6); and in those with levels c-1 mg-DST ≥ 5 µg/dL (OR = 1.2, 95% CI: 0.6–2.3, *p* = 0.654). The DST-related diagnostic accuracy was low, with areas under the receiver operating characteristic (ROC) curve < 0.61, when it came to predicting cardio-metabolic comorbidities in patients with AIs. Furthermore, the study findings showed that the c-1 mg-DST was inefficient in identifying/predicting people who already had cardio-metabolic comorbidities or who were going to develop them over time [60].

Additionally, no difference in the prevalence of T2DM between the group with MACS and the group without MACS was established according to another study (17.8% versus 15%, *p* = 0.246). There were no notable variations in the prevalence of T2DM in females, meaning 17.9% in the MACS sub-group versus 13.1% in the non-MACS sub-group (*p* = 0.278). However, males with MACS showed a statistically significantly increased rate of T2DM versus non-MACS: 24.7% (N = 93) versus 10% (N = 80), *p* = 0.012. In this analysis, T2DM was also linked, at least in women, to the prevalence of vertebral fractures, regardless of the bone mineral density (BMD) and the MACS confirmation. Hence, this indicates that female adults confirmed with T2DM and MACS should be rigorously monitored for bone fragility [52]. A total of 98 MACS patients (33 males and 65 females) were included in another retrospective analysis; in females with MACS, LDDST (0.5 mg/6-h for 48-h) and c-1 mg-DST were statistically significantly higher than in males with MACS. Moreover, a statistically significant correlation between ACS and fasting C-peptide, as well as the C-peptide-to-glucose ratio in females, was found by applying a logistic regression analysis. Overall, the prevalence of T2DM was similar in men versus women (33.3% versus 20%, *p* = 0.147) [57].

Three studies that analysed NFAI and MACS also included a control group. A cross-sectional study involving 58 patients with MACS, 89 with NFAI, and 64 controls was conducted by Rebelo et al. [53]; a similar prevalence of T2DM was found between the three groups (35.1% versus 37.1% versus 28.6%, *p* > 0.5) [53]. Through the analysis of their urine steroid profile, 73 participants (NFAI = 24, ACS = 25, and healthy controls = 24) in another cross-sectional study were able to identify changes. The study concluded that NFAIs appear to secrete a subtle, however, clinically relevant, excess of glucocorticoids. Furthermore, compared to patients with NFAIs or the control group, patients with ACS had a higher prevalence of T2DM (*p* = 0.003). T2DM was more common in patients with NFAI and UFC above the reference range (N = 5) than in those with NFAI and normal UFC levels (N = 19) (20% versus 0%, *p* = 0.046) [55]. In a prospective cohort on 601 individuals (331 males and 270 females) who had computed tomography scans (average age of 63.5 ± 14.4 years), 44 (7.3%) individuals had AI, and 20 patients (50%) did not supress the cortisol levels below 1.8 µg/dL. Notably, AI was associated with a higher risk of T2DM than the control group (31.8% versus 14.2%, *p* = 0.004), but no differences between the two groups of AIs, NFAI, and ACS (30% versus 35%, *p* = 0.74) were pinpointed. Multivariable regression analysis revealed a statistically significant correlation between the diagnosis of T2DM and the presence of AI (*p* = 0.003). Of note, a possible intrinsic, insufficient statistical power due to the low number of patients within each sub-group should be taken into consideration [65].

To conclude, while not all studies agreed on a higher ratio with concern to T2DM in MACS versus NFAIs, most data suggest a particular awareness with concern to this metabolic finding in MACS versus NFAI, but T2DM was identified, too, in up to 40% of the NFAIs sub-group in some studies. Further on, these results (mostly, based on retrospective or transversal study design) should be translated into individual management amid daily endocrine care [52,53,54,55,56,57,58,59,60,61,62,63,64,65,66,67].

### 2.2. Prevalence of Τ2DM in Patients with NFAI versus Controls (Patients without the Diagnosis of an Adrenal Adenoma)

Most notable is that even patients confirmed with NFAIs (who have c-1 mg-DST < 1.8 µg/dL) are more likely to acquire metabolic traits such as T2DM than controls (patients without any adrenal adenomas). Currently, there are conflicting data about the high frequency of impaired glucose metabolism in NFAIs, but recent findings (but not all) showed that the prevalence of T2DM in NFAI was higher than in controls without an adrenal lesion who were matched for other (general) cardio-metabolic risk factors. We identified four studies (one of cohort type, one retrospective and two of cross-sectional design) that compared NFAIs with controls; all these studies used the same cut-off for NFAI (c-1 mg-DST < 1.8 µg/dL) [68,69,70,71] (Table 2).

A total of 2791 patients were included, meaning 1243 subjects had NFAIs (female-to-male ratio of 1.11, a mean age of 60.42 years) and 1548 controls (female-to-male ratio of 0.91, an average age of 60.22 years). Across two confirmatory studies, patients with NFAI had a higher prevalence of T2DM than controls, while the other two did not confirm this specific aspect [68,69,70,71]. The highest T2DM prevalence was 45.2% in the NFAI group; however, compared to controls (45.2% versus 35%, *p* = 0.38), it was not a statistically significant difference (by applying ADA criteria from 2017 [72]). Because there were no differences in the groups’ age, gender, ethnicity, smoking status, use of statins, or history of atherosclerotic cardiovascular disease, the similarities in the groups’ glucose profiles may be explained [71]. According to another cohort study comparing adrenal adenomas with overt hormone excess (N = 1004) to age- and sex-matched participants without adrenal adenomas (N = 1004), the first sub-group displayed a statistically significant increased incidence of cardio-metabolic traits that included a higher unadjusted dysglycemia (43% versus 28%, *p* < 0.001) with an OR of 1.63 (95% CI: 1.33–2.00), dysglycemia being defined as a composite of prediabetes or diabetes. The prevalence of T2DM was higher in the adrenal adenoma group versus controls (27.5% versus 17.4%, *p* < 0.001). A potential bias was the fact that patients with MACS (c-1 mg-DST > 1.8 µg/dL) were included in the tumours’ sub-group, and no c-1 mg-DST was actually performed in 782 patients [69]. Another cohort of 154 NFAI patients and a 1:3 age- and gender-matched control group (N = 462) revealed a statistically significant increased OR for diabetes in the NFAI group (OR = 1.89, 95% CI: 1.17–3.06). In the NFAI group, the prevalence of T2DM was higher compared to the control group (25.3% versus 14.5%, *p* < 0.05). Also, FPG and HbA1c were found higher in NFAI versus controls (108 ± 26.5 versus 99.5 ± 17.7 mg/dL, *p* < 0.001, respectively, 6.1 ± 0.9 versus 5.9 ± 0.6%, *p* = 0.009) [70]. On the other hand, another study showed a T2DM prevalence that was comparable between NFAI (N = 43) and healthy controls (N = 42) (27.9% versus 26.1%, *p* = 0.209). Interestingly, T2DM was statistically significantly correlated with masked hypertension (OR = 2.07, *p* = 0.044) [68].

### 2.3. Prevalence of T2DM in Patients Diagnosed with NFAI versus Possible ACS versus ACS

According to the criteria released in 2016 by ESE, a possible ACS was defined in AI patients with c-1 mg-DST between 1.8 and 5 µg/dL [73]. However, recent recommendations showed that all people with AI and a c-1 mg-DST > 1.8 µg/dL are classified as MACS, thus eliminating the distinction between possible ACS and ACS [23]. Noting this subtle change of terminology and definition, we distinctly analysed the studies addressing prior terms/definitions (namely, “possible” ACS). These five studies using the same cut-off to characterize the adrenal tumours included NFAI (c-1 mg-DST < 1.8 µg/dL), possible ACS (c-1 mg-DST between 1.8 and 5 µg/dL), and ACS (c-1 mg-DST > 5 µg/dL). Almost one-third (between 10.9 and 37%) of all AI has a possible ACS, while T2DM prevalence varied from 20.6% to 33.3% among this specific sub-group [74,75,76,77,78]. For instance, the prevalence of T2DM increased from NFAI to possible ACS and ACS (18.2%, 23%, and 26.7%, respectively, *p* < 0.001) in a multicentre retrospective cohort that primarily investigated the impact of cortisol secretion autonomy on mortality in AI (inclusion criteria were age ≥ 18 years, uni- or bilateral AIs with a diameter ≥ 1 cm detected by cross-sectional imaging). There were 3656 patients with AI, meaning NFAI (N = 2089), possible ACS (N = 1320), and ACS (N = 247), 64% of the entire cohort being females (a median age of 61 years and a median follow-up of 7 years). There was a statistically significant increase in all-cause mortality (adjusted for age, sex, comorbidities, and prior cardiovascular events) in the sub-group diagnosed with possible ACS [hazard ratio (HR) of 1.52 (95% CI: 1.19–1.94)] and ACS [HR of 1.77 (95% CI: 1.20–2.62)]. The rate of mortality in the possible ACS and ACS groups was higher than in the NFAI group (12.7% versus 16.6% versus 6.8%). The confirmation of ACS was associated with a 4-fold increase in adjusted death in females under the age of 65, while in males, this aspect was not confirmed [74].

In another study, patients with NFAI had a T2DM prevalence of 15.4% versus the sub-group with possible ACS (20.6%, *p* = 0.606), and the HbA1c profile was similar between these two sub-categories (5.76 ± 0.83% versus 5.82 ± 0.57%, *p* = 0.802), while FPG in NFAI patients (5.50 ± 0.75 mmol/L) and possible ACS (5.55 ± 0.67 mmol/L) were statistically significantly higher than seen in healthy controls (4.93 ± 0.39 mmol/L, *p* = 0.003, respectively, *p* < 0.001). This was an observational case–control analysis of a small sample, including 26 patients with NFAI, 34 subjects with possible ACS, and another 32 age-matched healthy controls without any adrenal tumours. The average age of the AI patients was 57.47 ± 7.17 years (47 females and 13 males), and the primary endpoint was to screen for depression and evaluate the quality of life in patients with NFAI and possible ACS. A midnight salivary cortisol of 86.95 nmol/L had a high specificity (80%) as well as high sensitivity (82.8%) to identifying mild depression in individuals with possible ACS, according to the ROC curve [75].

Across two studies [76,77], a distinct group of patients diagnosed with CS was analysed with respect to the T2DM rate, and specific data were provided (while another only provided a general rate of 28.3% in patients confirmed with AIs [78]). Paradoxically, the T2DM prevalence was higher in patients with possible MACS and MACS versus CS. In one study, 1305 people with benign adrenal tumours were included, and according to their clinical presentation and c-1 mg-DST results, the participants were divided into four sub-groups: NFAI (N = 649, female-to-male ratio of 416 to 233, median age of 58 years), possible MACS (N = 451, female-to-male ratio of 303 to 148, median age of 64 years), MACS (N = 140, female-to-male ratio of 103 to 37, median age of 63 years), and adrenal CS (N = 65, female-to-male ratio of 56 to 9, median age of 48 years). The frequency of T2DM was higher in subjects with possible MACS and MACS than in adrenal CS (32.2% versus 33.7% versus 31.5%). Diabetic individuals with CS and MACS had a higher rate of insulin therapy need than patients with NFAI [adjusted prevalence ratio (aPR) of 3.06 (95% CI: 1.60–5.85); respectively, aPR of 1.89 (95% CI: 1.01–3.52)] [76]. Similarly, another cohort pinpointed that T2DM prevalence in NFAI and possible ACS sub-groups (of 24.2%, respectively, 33.3%) was higher than found in CS (of 13%) with borderline statistical significance (*p* = 0.09) [77]. Of note, different editions of ADA criteria were used amid these mentioned studies [79,80,81] (Table 3).

### 2.4. Analysing the Prevalence of T2DM in CS Sub-Groups Amid the Studies That Included Subjects Confirmed with NFAIs

A sub-group of individuals with CS was included in four studies, with rather unexpected results (two of the studies were mentioned above, showing a higher T2DM rate in patients with possible MACS and MACS versus CS [76,77]). Also, Delivanis et al. [82] found that individuals with MACS presented a higher rate of T2DM or IFG, in contrast to those with confirmed CS (45% versus 35%) [82]. In another prospective analysis, 213 people identified with different types of adrenal adenomas (22 with CS, 92 with MACS, and 99 with NFAI) showed a similar T2DM prevalence (41% versus 41% versus 41%, *p* = 0.99) [83] (Table 4).

### 2.5. Prediabetes (IFG and/or IGT) in Patients with NFAI and MACS

We identified four studies that provided data on the general category of the glucose profile, specifically, prediabetes, and three of them have already been mentioned for introducing data according to other subsections across our methods [56,69,70]. The Homeostasis Model Assessment-Insulin Resistance (HOMA-IR) formula was used to quantify the insulin resistance, which is generally confirmed if HOMA-IR is ≥2.5 [84]. Of note, this represents a very useful mathematical model, highlighting insulin resistance in daily practice, being considered the most common method of assessing insulin sensitivity, which is easy to use and only requires a fasting blood sample; notably, different lab standards and cut-offs varied over the time in normal subjects. Szychlinsk et al. [85] did not reveal a higher HOMA-IR value when comparing the NFAI to the control group (that was matched for age, gender, and BMI); of note, individuals with a history of T2DM were excluded. The fasting insulin level of the NFAI group was statistically significantly increased versus controls (11.4 ± 4.9 versus 8.9 ± 5.8, *p* = 0.03). The association between the cortisol value (c-1 mg-DST) and fasting insulin showed borderline significance (r = 0.254; *p* = 0.08). The NFAI group experienced IGT more often compared to controls (27% versus 0.7%, *p* = 0.026). No association between the carotid intima-media thickness and lipid levels, FPG, or glucose level in the 2 h OGTT was identified [85]. However, Kim et al. [70] confirmed that in the NFAI group, the HOMA-IR score was statistically significantly higher than controls (2.80 ± 2.17 versus 2.00 ± 1.10, *p* = 0.022). Insulin resistance was 2-fold more likely to occur in patients confirmed with NFAI versus controls (OR = 2.03, 95% CI: 1.06–3.90) [70]. Adamska et al. [56] showed that prediabetes prevalence was similar in NFAI and MACS (34.3% versus 26.8%, *p* < 0.35). Impaired fasting glucose (IFG), defined as a FPG value between 100 and 125 mg/dL, and/or IGT, defined at a serum glucose level of 140–199 mg/dL in the 2 h OGTT, were considered to be indicators of prediabetes in this cohort [56]. Another previously mentioned cohort study including patients (N = 1004) without overt hormone excess (including 81 patients with MACS; of note, 782 patients had no data on c-1 mg-DST) and age- and sex-matched controls without adrenal adenomas (N = 1004) revealed that prediabetes was statistically significantly more common in the tumour group than controls (15.4% versus 10.5%, *p* < 0.001), prediabetes being defined by a HbA1c value between 5.7% and 6.4% [69] (Table 5).

### 2.6. Longitudinal Data with Respect to the Glucose Profile in Patients with NFAI and MACS

It is still debatable which should be the exact protocol of long-term follow-up with respect to NFAIs or how closely they need to be monitored regarding the cardio-metabolic traits, not only the specific endocrine and adrenal imagery panel [86]. Few and rather equivocal longitudinal studies have been conducted on the glucose profile in AI patients. We identified eight such studies [52,54,59,62,69,70,86,87], and six of them were discussed earlier [52,54,59,62,69,70]. The longest period of surveillance was 10.5 years; this was a study on 67 participants (70.1% females and 29.9% males) with a mean age of 57.9 years throughout a prospective study design. All patients were diagnosed with NFAI at the time of first presentation (c-1 mg DST ≤ 1.8 µg/dL), and there was a statistically significant increase in the rate of T2DM during follow-up (0.03% at baseline versus 17.9% during follow-up, *p* = 0.002) [87]. In research published by Araujo-Castro et al. [59], 10.5% of NFAIs developed ACS after a mean follow-up period of 41.3 months; this was associated with higher (initial) serum cortisol levels following the DST. Moreover, c-1 mg-DST on first diagnosis was statically significantly higher in patients with T2DM versus without this disease (1.28 ± 0.36 versus 1.14 ± 0.37 µg/dL, *p* = 0.004). During follow-up, 53 patients acquired one or more additional comorbidities, and 5.7% of these individuals had T2DM [59]. Another study showed that long-term monitoring is necessary for patients with NFAIs; the control group was followed for 7.2 years and the patients with adrenal adenomas for a median of 6.8 years, and they had a higher unadjusted 10-year cumulative incidence of new metabolic and cardiovascular diseases, including dysglycemia, than controls (18% versus 14%) as well as a higher unadjusted overall mortality than controls (15% versus 10% at 5 years, 28% versus 21% at 10 years, and 41% versus 35% at 15 years; *p* < 0.001). After adjusting for the baseline smoking status and cardio-metabolic traits, the overall mortality between the two groups was similar during surveillance. Cardio-metabolic outcomes in patients with NFAI (N = 141) versus MACS (N = 81) showed a higher unadjusted overall mortality in MACS compared to NFAIs (3% versus 2% at 5 years, 20% versus 9% at 10 years, and 37% versus 19% at 15 years), with an adjusted HR of 2.01 (95% CI: 0.92–4.41) for age, sex, BMI, smoking, and the prevalence of cardio-metabolic risk factors [69].

Another long-term cohort (154 subjects with NFAIs versus a 1:3 age- and gender-matched control group of 462 individuals) revealed a statistically significantly increased OR for diabetes in the NFAI group (OR = 1.89, 95% CI: 1.17–3.06), while during follow-up, persons with NFAI and T2DM had larger adrenal lesions than those without T2DM (*p* = 0.048) [70]. Another longitudinal study (follow-up of 31.4 months) pinpointed no changes in the T2DM prevalence with concern to the sub-groups that experienced AI enlargement and those that did not (15.1% versus 20.2%, *p* = 0.389). It was found that patients who experienced ACS during follow-up had a higher rate of IFG than those who did not (6.9% versus 42.9%, *p* = 0.001). Overall, there was no additional statistically significant change in the glucose metabolism during the follow-up [62]. Another mentioned retrospective analysis identified 478 patients with NFAI and 231 patients with ACS; T2DM was diagnosed in 24.3% of all patients; a median of follow-up of 28 months (interquartile range was between 2.0 and 125.3) revealed that 24/709 patients were newly confirmed with T2DM; overall, there were no differences between the two sub-groups regarding this new ailment (HR of 1.17, 95% CI: 0.52–2.64) [54]. Furthermore, in the longitudinal arm (N = 126 patients with AIs, follow-up of at least 24 months), the T2DM prevalence was similar for MACS versus non-MACS (15.9% versus 20%, *p* = 0.221) [52]. In 80 subjects with NFAI patients and pancreato-steatosis, Candemir et al. [88] identified more anomalies of the glucose metabolism during follow-up (versus controls). The pancreatic and hepatic lesions at baseline were assessed via non-contrast abdominal computed tomography. Following two years of surveillance, FPG changes in the adenoma group were statistically significantly higher compared to the control group (*p* = 0.002), as seen with HbA1c levels (*p* < 0.001) [88] (Table 6).

### 2.7. Adrenalectomy versus Conservative Management in MACS: The Impact on Glucose Profile (T2DM Prevalence)

The optimum treatment strategy for MACS is still an open issue; the main options are adrenalectomy versus a conservative approach (including the medical treatment of MACS-associated comorbidities such as diabetes, hypertension, etc., regales the decision for surgery). The level of evidence is contradictory, although across six trials including 1039 individuals (674 subjects underwent adrenalectomy and another 365 were followed under conservative care), most of them found that the MACS-related glycaemic profile improved post-operatively (but not all studies agreed) [89,90,91,92,93,94].

Morelli et al. [89] included 55 adults aged between 40 and 75 years who had AI larger than 1 cm and c-1 mg-DST between 1.8 and 5 µg/dL; they were randomized into two groups: conservative (female-to-male ratio of 4, mean age of 66.1 ± 9.1 years) or adrenalectomy (female-to-male ratio of 2.1; average age of 62.5 ± 10.4 years). A 6-month follow-up revealed that the gluco-metabolic improvement was more pronounced after surgery (28% versus 3.3%, *p* = 0.02) [89]. Similarly, adrenalectomy was performed on 117 subjects confirmed with subclinical CS versus 18 people in the non-operative group, according to another cohort; T2DM was less common in the surgical than in the conservative group (27% versus 55.6%) during follow-up (*p* = 0.0255) [90].

On the other hand, Thompson et al. [91] showed no variations in the prevalence of T2DM or other morbidities between patients with ACS and NFAI, while during follow-up, after adrenalectomy, the need for antidiabetic drugs was similar in patients with ACS who were under medical therapy (*p* > 0.5) [91]. Another study included a total of 260 patients with AI (61 patients were enrolled in the surgical group and another 199 represented the conservative group). Within the first group, T2DM prevalence was stationary following the adrenal surgery (18% at diagnosis versus 26.2% at follow-up, *p* > 0.5), as seen in the other group (21.1% at baseline versus 30.2% at follow-up, *p* > 0.5). T2DM was less common in the non-operated patients with NFAI, as compared to possible ACS and ACS sub-groups according to the follow-up assessments (23.8% versus 35.6% versus 40.0%; *p* < 0.01). Within the sub-groups with ACS, possible ACS, and NFAI, the rate of using antidiabetic medication was similar [92]. A retrospective cohort of 171 individuals with AI who underwent laparoscopic adrenalectomy and had normal hormone levels at baseline and hypertension showed that 76% (N = 130) of the patients had a remission of high blood pressure, and 24% (N = 41) of them had persistent hypertension. T2DM was similar in both groups, with or without hypertension remission (8.5% versus 17.1%, *p* = 0.202) [93].

A complex study firstly revealed through cross-sectional analysis that MACS patients exhibited a substantially higher T2DM prevalence at baseline than the NFAI group (19% versus 7%, *p* < 0.05). A higher cortisol level in MACS was linked to a more severe gluco-metabolic and lipid profile. Hence, patients with MACS (c-1 mg DST > 1.8 µg/dL) who also had cardiovascular co-morbidities (such as obesity, metabolic syndrome, arterial hypertension, glucose intolerance, or T2DM, dyslipidaemia, and obesity) had similar rates of major cardiovascular events and mortality as those who displayed a level of c-1 mg DST above the cut-off of 5.0 µg/dL. The longitudinal analysis showed that T2DM prevalence in the operative group before adrenalectomy (20.7%) statistically significantly decreased after adrenalectomy (7.4%, *p* < 0.05), while the non-operative group remained stationary at the end of the follow-up period versus initial assays (26.7% versus 23.5%, *p* > 0.5) [94] (Table 7).

## 3. Discussion

In light of the rising incidence of NFAI/AIs due to easier access to imagery scans and endocrine evaluation across the spectrum of modern medicine, it is critical to assess if these patients have an increased frequency of cardio-metabolic disorders that worsen their overall comorbidity and mortality profile, including via the diagnosis of T2DM in MACS and (non-MACS) NFAIs. Growing statistical evidence connected the ACS/MACS sub-group to a high-risk category amid these concerns. However, the extended category of NFAIs was already found to be associated with at least a two-fold increased risk of T2DM and higher FPG levels versus controls [95,96]. The exact influence of cortisol secretion on the glucose profile is still an open issue in NFAI/MACS, and, most probably, there are currently two main biases; one is the fact that the glycaemic status is under the influence of a multifactorial panel of elements, such as age, menopausal status, association with obesity, vitamin D deficiency, and inflammatory status and a genetic/epigenetic influence, on the one hand, and, on the other, the specific assessments of the cortisol profile in these adrenal tumours varied over the years and, while DST remains the gold standard, this testing still represents a less-than-perfect tool to truly capture the endocrine and metabolic essence of these adrenal masses. To our knowledge, this comprehensive review included one of the largest recent analyses according to our methods in the field of the glucose profile in NFAIs/MACS. We analysed 37 studies with various study designs (14 retrospective studies and another 13 cross-sectional studies as well as 4 cohorts, 3 prospective, and 2 case–control studies) according to distinct endpoints that included a total of 17,391 individuals with a female-to-male ratio of 1.47 (aged between 14 and 96 years) [52,53,54,55,56,57,58,59,60,61,62,63,64,65,66,67,68,69,70,71,74,75,76,77,78,82,83,85,87,88,89,90,91,92,93,94]. Whether this timeframe of search amid the recent COVID-19 pandemic and early post-pandemic years might have influenced these data it still a matter of debate, as seen in other medical areas [97,98,99].

### 3.1. Pathogenic Considerations of Glucose Profile Anomalies in NFAIs/MACS

Persistently high circulating insulin levels (hyperinsulinemia) are usually associated with obesity and T2DM. Subclinical adrenal cortex activity may cause a vicious cycle, whereby hypercortisolemia leads to an increased insulin resistance, which, in turn, leads to hyperinsulinemia that has negative effects, including the onset of T2DM. It is unclear if primary insulin resistance and compensatory hyperinsulinemia lead to adrenal lesions, or if insulin resistance is only a by-product of AI-related slightly elevated cortisol [100,101]. Insulin represents an anabolic hormone that promotes growth together with insulin-like growth factor 1 (IGF-1). The biological actions of IGF1 are transduced by a group of transmembrane receptors that act on various organs. The normal adrenal cortex contains insulin receptors (IRs) such as IGF-1R, and IGF-2 receptors/mannose-6-phosphate (M6P) that are essential for the activation of various other adrenal gland receptors that, further on, promote growth and development. Insulin binds to IGF-1R and IRs to initiate subsequent pathways including phosphoinositide 3-kinases (PI3K) and mitogen-activated protein kinases (MAPK), so one theory is that high insulin levels by binding to IGF-1R may promote cell proliferation, including at the level of the adrenal glands as seen in others sites [102,103,104,105]. Furthermore, it has been shown that increased IGF-2 levels and IGF-1R overexpression are frequently associated with the confirmation of adrenocortical tumours [106]; even a correlation between insulin resistance and NFAIs has been reported [107] between the severity of the insulin resistance and the size of the adrenal mass [108]. Alternatively, we mention that the presence of T2DM might increase the risk of certain neoplasia with different origins [109,110,111] and, potentially, of the cortico-adrenal tumours [112]. Among the underlying mechanisms, the hyperglycaemia-induced mitogenic pathways and the use of insulin analogues [113,114,115] are added to hypercortisolism-associated insulin resistance [116]. Moreover, in the MACS sub-group, the anomaly of the adrenal hormones is more obvious than in the general category of NFAIs, while tumour removal suggested an improvement in the glucose profile anomalies as therapeutic evidence [117,118].

### 3.2. Cortisol Excess and Signal Transduction Pathways Amid Glucose Metabolism: Dual Interplay

Tumours with ACS/MACS are essentially related to the cortisol-related effects on the glucose profile that are expected to be less obvious than seen in typical CS, but, as mentioned, the clinical evidence strongly suggests a tight connection with the glycaemic metabolism damage. A long-term excessive amount of cortisol is traditionally known to be involved in the metabolic pathways that regulate this metabolism, including reduced pancreatic insulin secretion and increased hepatic gluconeogenesis due to the recruitment of various enzymes (phosphoenolpyruvate carboxykinase-1, glucose-6-phosphate dehydrogenase), decreased glucose uptake by adipocytes mediated by adipokines, and increased lipolysis [119,120,121]. The indirect effects of hypercortisolism include long-standing hyperglycaemia, which results in oxidative damage within the cell and insulin resistance. Oxidative stress is defined as an imbalanced redox state, in which antioxidant cellular mechanisms are impaired and reactive oxygen species (ROS) are excessively produced and accumulated. As free radicals, ROS interact with lipids, proteins, and nucleic acids (deoxyribonucleic acid, ribonucleic acid) by using their available electrons, thus causing protein glycation and oxidative degeneration, which induces cellular damage and affects pancreatic β cells, hence, insulin synthesis. HbA1c (as mentioned in some of the cited papers) and fructosamine levels are two indicators that are used to measure the degree of such protein glycation in daily practice [122,123,124,125]. On the other hand, long-term hyperglycaemia may lead to a glucocorticoid-resistant status in the hypothalamus–pituitary–adrenal axis by increasing the ROS and oxidative stress, which may damage the glucocorticoid receptor function that further blunts the axis response and causes reactive hypercortisolism in T2DM [126,127].

### 3.3. Genetic Insights between Cortisol and Glucose Crossroads

Following their synthesis and secretion, glucocorticoids are metabolized by 11β-Hydroxysteroid Dehydrogenase (11βHSD), which controls the peripheral interconversion between cortisone and cortisol [128]. This enzyme has two isoforms that are presented in different tissues [129]; 11βHSD type 1 enzyme has a pronounced metabolic role, and altering 11βHSD1 activity in T2DM patients improves diabetes control [130,131]. Secondarily, glucocorticoids mediate their effects through glucocorticoid receptors (encoded by *NR3C1* gene); variations in the sensitivity to endogenous glucocorticoids seem strongly correlated with the *NR3C1* polymorphism [130,132,133]. Some polymorphisms have been connected to a higher risk of metabolic traits, including T2DM [130,134,135], such as N363S polymorphism [136], while the ER22/23EK polymorphism was associated with reduced first-phase glucose-stimulated insulin secretion [137]. Interestingly, increased glucocorticoid sensitivity caused by N363S or Bcll polymorphisms was associated with the most severe lipids and gluco-metabolic profile [138]; moreover, some BclI variants have been connected to AI development [139,140,141,142,143]. These complex factors should be taken into consideration when analysing why some patients with NFAIs/MACS seem prone to develop glycaemic anomalies ranging from IGT to overt diabetes associated with multidisciplinary complications and why the results from clinical trials are not homogenous in different populations.

### 3.4. Accuracy of the Endocrine Tests in NFAI/MACS

The accuracy of c-1 mg-DST in order to predict prevalent or incident comorbidities in patients with AI/NFAIs is still a matter of debate. c-1 mg-DST represents the commonly accepted standard test for the diagnosis of NFAI (c-1 mg-DST < 1.8 µg/dL) and MACS (usually defined as associating a c-1 mg-DST > 1.8 µg/dL). Other studies supported the idea that in euglycemic and normotensive subjects with AIs, an LDDST value of >1 and >1.37 µg/dL had a higher predictive value with respect to insulin resistance and cardiovascular risk [116]. Interestingly, c-1 mg-DST, with a cut-off value of 1.19 µg/dL, might exclude post-adrenalectomy adrenal insufficiency as indirect evidence of pre-surgical ACS [144]. Alternatively, a lower cut-off level of 0.9 µg/dL may work as a good discriminator in identifying subjects with at least one prevalent (osteoporotic) fragility fracture, hypertension, or T2DM [145]. In obese individuals with a normal cortisol response (c-1 mg-DST < 1.8 µg/dL), the presence of hypertension and T2DM may be independently predicted by the actual level of c-1 mg-DST [146]. Steroid profiling using mass spectrometry represents a different approach that is currently less feasible in daily practice [147]. Further testing to confirm the diagnosis of MACS includes measuring UFC, serum baseline ACTH (a cut-off < 10 pg/mL is very suggestive), and dehydroepiandrosterone sulfate (a cut-off < 40 µg/dL might help the diagnosis) [148,149,150].

### 3.5. Integrating the Glucose Status to the Panel of Complications in NFAI/MACS

Even a mild excess of glucocorticoids may be associated with numerous cardio-metabolic comorbidities, as well as digestive, pancreatic, hepatic, osseous, neurologic, dermatologic, infectious involvements, etc. [151,152,153]. Patients confirmed with MACS had a higher risk than those with NFAI and healthy controls to experience arterial hypertension and cardiovascular events [154,155,156,157,158]. Furthermore, as shown, even individuals with NFAI are at increased risk of cardiovascular events than the general population [159]. In MACS, there was a higher incidence of cardiovascular events during follow-up amid some longitudinal studies of more than a decade [160]. Among the pathogenic elements of these cardio-metabolic elements, new emergent biomarkers are mandatory, as recently proven by a higher level of pentraxin 3, an acute-phase glycoprotein that has been involved in this particular instance of adrenal tumour-related cardio-metabolic traits [161]. Moreover, a potential elevated risk of thromboembolic events was reported in MACS [162]. As mentioned, people with MACS have a noticeably increased rate of osteoporosis and vertebral fractures compared to adults diagnosed with NFAI [163,164,165,166] (and adrenalectomy might reduce the risk of vertebral fractures [167]) and even of sarcopenia [168,169]. Overall, a higher mortality should be kept in mind (primarily through cardiovascular disease) that might elevate the tumour- and T2DM-related burden, as seen in other endocrine and non-endocrine tumours [170,171,172,173,174].

### 3.6. T2DM Management in Subjects with MACS/NFAIs

As previously specified, the main issue remains the radical decision in individuals confirmed with MACS and even NFAI, and this decision should take into consideration a large panel of clinical elements/co-morbidities (including T2DM), in addition to the hormonal and imagery assays. Adrenalectomy seems to be beneficial from the cardio-metabolic perspective in MACS [175,176,177], with the laparoscopic approach being preferred [178]. Medical therapy such as mifepristone, a glucocorticoid receptor blocker, may improve the insulin resistance and even provide a bridge to surgery [179]. However, considering the potential adverse effects and a rather increased expense of a likely lifelong therapy, there is currently very little evidence to justify the long-term use of the medical treatment in patients with MACS. Currently, there are no specific guidelines to address managing T2DM in subjects confirmed with MACS, hence the importance of addressing the topic. Therefore, it is suggested to take into account the general step-by-step process, as follows: managing and treating increased glucose levels with carefully selected antidiabetic drugs [180]. As a long-term backup plan to adrenalectomy, patients with MACS-T2DM may benefit from therapy with drugs that improve insulin resistance such as metformin [181]. This biguanide increases insulin sensitivity and in cases of hypercortisolemia, whereas a significant number of patients have both hyperglycaemia and insulin resistance, so this might be helpful as a first-line therapy. Moreover, metformin suppresses tumour growth through a variety of pathways, which may be beneficial with regard to the development of adrenal adenomas [182]. In individuals with MACS, the normalization of hypercortisolism remains the first step toward achieving metabolic control. Since there are only limited long-term studies evaluating the cardio-metabolic outcomes in MACS, further large multicentre prospective trials are necessary.

Overall, exploring the glucose profile in this specific type of endocrine (adrenal) tumour, including the sub-categories with hormonal excess (even mild cortisol over-production), needs further larger studies, particularly of interventional and longitudinal designs. Moreover, the identification of the best strategy with respect to the long-term approach in the matter of glycemic features is still an open question, as part of the diagnosis at initial tumour identification, as part of the long-standing screening protocols following patients with incidentalomas who are not referred to surgery, and as being a potential contributor to the multidisciplinary management of the adrenal cortex neoplasia. We need multifactorial (stratified) models that should take into consideration the specific cardiovascular risk as well as other (potential) co-morbidities, such as reduced bone mineral density and the deterioration of bone microarchitecture, mood changes, depression, and overall quality of life, in addition to glucose-related traits.

In terms of the limitations in the current work, we mention that this is the result of a single database search across a non-systematic review, which was preferred to a systematic review in order to achieve a more flexible approach and cover more data that came from studies with different designs and various sample sizes and protocols. Yet, to our knowledge, this is one of the most complex sample-based analyses in this specific domain according to the most recent five years of publications.

## 4. Conclusions

To our knowledge, this large and complex comprehensive review covered the most important aspects according to the recent publications in the field of MACS/NFAI-related T2DM across our methods (n = 37 studies, N = 17,391 individuals, aged between 14 and 96 years). The main take-home messages are listed as follows:Most Data Came from Retrospective (n = 14) and Cross-Sectional Studies (n = 13), and Only Three Prospective Studies Were Identified.The Analysed Population Showed that the Female Population Was Slightly More Disposed (the Female-to-male ratio of 1.47), but the Results in the Glucose Profile Did Not Show a Specific Gender-Related Burden in These Adults.The MACS Prevalence Amid NFAIs Was 10 to 30%. Most Studies Sustained a Higher T2DM Prevalence in MACS (12 to 44%) versus NFAIs.However, a Few Studies also Showed a Similar Rate in NFAIs (up to 45%) and MACS or a Higher T2DM Rate than Seen in Healthy Controls.Prediabetes (such as IFG or IGT) May Be More Frequent in MACS versus NFAIs or NFAIs versus Controls (No Homogenous Results).Four Studies Introduced a CS Sub-Group, and, Paradoxically, Only Half of Them Confirmed a More Severe Glucose Profile versus MACS/NFAIs.The Longest Period of Follow-Up with Concern to the Glycaemic Profile Was 10.5 Years, and One Cohort Showed a Significant Increase in the T2DM Rate at 17.9% Compared to the Baseline Value of 0.03%.Inconsistent Data Coming from Six Studies Enrolling 1039 Individuals that Underwent Adrenalectomy (N = 674) and Conservative Management (N = 365) Pinpointed the Impact of Adrenalectomy in NFAIs that Improved the Regulation of the Glucose Metabolism after Adrenalectomy versus Baseline versus Conservative Management (n = 3).Noting These Data, Awareness of the T2DM and Other Glucose Profile Anomalies in NFAIs/MACS Represents the Key Factor in the Transversal and Longitudinal Approach.

## Figures and Tables

**Figure 1 biomedicines-12-01606-f001:**
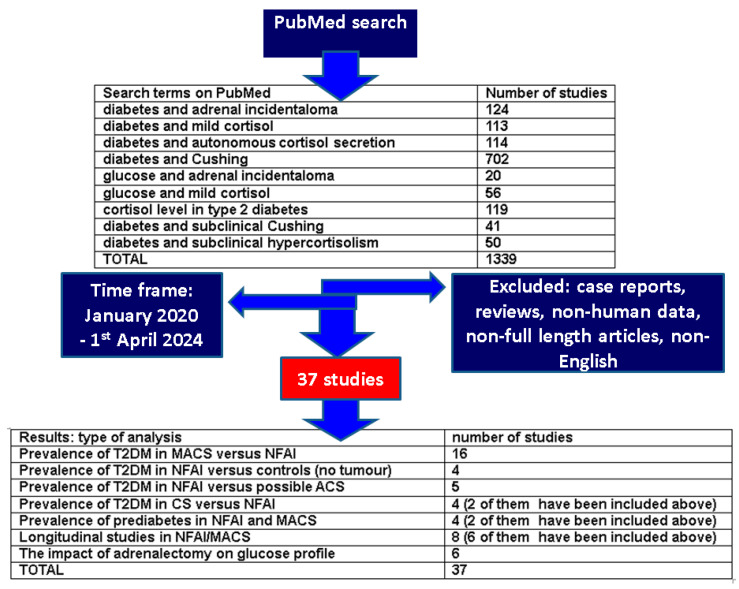
Flow chart diagram of search and main sections of results according to our methods.

**Table 1 biomedicines-12-01606-t001:** Summary of the studies showing T2DM prevalence in MACS versus (non-MACS) NFAIs according to our methods (the display starts with the most recent publication date) [52,53,54,55,56,57,58,59,60,61,62,63,64,65,66,67].

First Author Year of Publication Reference Study Design	Studied Population Number of Patients Age (Years) Gender (F/M)	Criteria for the Diagnosis of T2DM	The Prevalence of T2DM	Glucose Parameters [(Mean ± SD or Median (IQR)]
Favero 2024 [52] Cross-sectional study	**N = 444 AI** F/M = 271/173 Mean age = 61.8 ± 11.5 y (range: 21–89) y **MACS** = 230 (c-1 mg-DST > 1.8 µg/dL) F/M = 137/93 Mean age = 63.6 ± 9.5 y (range: 24–83) y **NFAI** = 214 (c-1 mg-DST ≤ 1.8 µg/dL µg/dL) F/M = 134/80 Mean age = 59.9 ± 13.0 y (range: 21–89) y	FPG ≥ 126 mg/dL 2 h-plasma glucose ≥ 200 mg/dL during OGTT Symptoms of diabetes plus casual plasma glucose concentration plasma glucose ≥ 200 mg/dL (ADA 2003)	**AI:** T2DM = 16.4% **MACS:** T2DM = 17.8% **NFAI:** T2DM = 15% **T2DM**: MACS vs. NFAI, *p* = 0.246	NA
Rebelo 2023 [53] Cross-sectional study	**N = 211** patients with or without adrenal adenomas **AI** = 147 (ACS + NFAI) F/M = 122/25 Mean age = 63.3 ± 9.9 y **ACS** = 58 (c-1 mg-DST > 1.8 µg/dL) F/M = 50/8 Mean age = 64.3 ± 9.8 y **NFAI** = 89 (c-1 mg-DST < 1.8 µg/dL) F/M = 72/17 Mean age = 62.7 ± 9.9 y **Controls** = 64 without adrenal tumours F/M = 46/18 Mean age = 60.5 ± 10.8 y	FPG ≥ 126 mg/dL 2 h-plasma glucose ≥ 200 mg/dL during OGTT HbA1c ≥ 6.5% in a patient with classic symptoms of hyperglycaemia or hyperglycaemic crisis, a random plasma glucose ≥ 200 mg/dL (ADA 2021) Dysglycemia included: T2DM, impaired fasting glucose, and impaired glucose tolerance.	**AI:** T2DM = 36.3% Dysglycemia = 82.1% **ACS:** T2DM = 35.1% Dysglycemia = 80.4% **NFAI:** T2DM = 37.1% Dysglycemia = 83.1% **Controls** T2DM = 28.6% Dysglycemia = 67.7% **T2DM:** AI vs. controls, *p* = 0.28 ACS vs. controls, *p* = 0.54 **Dysglycemia:** AI vs. controls, *p* = 0.02 ACS vs. controls, *p* = 0.07	**AI:** FPG = 5.93 (3.82–12.65) mmol/L HbA1c = 5.9 (4.3–10.5)% **ACS**: FPG = 5.93 (3.8–12.6) mmol/L HbA1c = 5.8 (4.4–8.9)% **NFAI:** FPG = 5.93 (3.9–12.2) mmol/L HbA1c = 6 (4.3–10.5)% **Controls:** FPG = 5.49 (3.60–8.99) mmol/L HbA1c = 5.9 (4.9–8.8)% **FPG:** AI vs. controls, *p* = 0.04 ACS vs. controls, *p* = 0.12 **HbA1c:** AI vs. controls, *p* = 0.55 ACS vs. controls, *p* = 0.72
Brox-Torrecilla 2023 [54] Retrospective study	**N = 709 AI** F/M = 397/312 Mean age = 63.4 ± 10.8 y **ACS** = 231 (c-1 mg-DST ≥ 1.8 µg/dL) F/M = 131/100 Mean age = 65.0 ± 10.60 y **NFAI** = 478 (c-1 mg-DST < 1.8 µg/dL) F/M = 267/211 Mean age = 62.7 ± 10.77 y	FPG ≥ 126 mg/dL 2 h-plasma glucose ≥ 200 mg/dL during OGTT HbA1c ≥ 6.5% In a patient with classic symptoms of hyperglycaemia or hyperglycaemic crisis, a random plasma glucose ≥ 200 mg/dL (ADA 2022)	**ACS:** T2DM = 27.7% T2DM and HbA1c > 7% = 35.4% T2DM and HbA1c > 8% = 14.6% **NFAI:** T2DM = 22.6% T2DM and HbA1c > 7% = 37.8% T2DM and HbA1c > 8% = 9.5% **T2DM:** ACS vs. NFAI, *p* = 0.137 **T2DM and HbA1c > 7%:** ACS vs. NFAI, *p* = 0.787 **T2DM and HbA1c > 8%:** ACS vs. NFAI, *p* = 0.386	**ACS:** FPG = 112.3 ± 35.56 mg/dL HbA1c = 6.5 ± 1.36% **NFAI:** FPG = 105.0 ± 29.05 mg/dL HbA1c = 6.1 ± 0.89% **FPG:** ACS vs. NFAI, *p* = 0.004 **HbA1c:** ACS vs. NFAI, *p* = 0.005
Araujo-Castro 2023 [55] Cross-sectional study	**N = 73** patients with or without adrenal adenomas **ACS** = 25 (c-1 mg-DST > 1.8 µg/dL) F/M = 16/9 Mean age = 70.2 ± 7.83 y **NFAI** = 24 (c-1 mg-DST ≤ 1.8 µg/dL) F/M = 17/7 Mean age = 67.4 ± 9.68 y **Controls** = 24 (without adrenal tumours) F/M = 18/6 Mean age = 65.5 ± 9.63 y	FPG ≥ 126 mg/dL 2 h-plasma glucose ≥ 200 mg/dL during OGTT HbA1c ≥ 6.5% In a patient with classic symptoms of hyperglycaemia or hyperglycaemic crisis, a random plasma glucose ≥ 200 mg/dL (ADA 2018)	**ACS:** T2DM = 44% **NFAI:** T2DM = 4.22% **Controls:** T2DM = 20.8% **T2DM:** ACS vs. NFAI vs. controls, *p* = 0.003	**ACS:** FPG = 104.8 ± 22.74 mg/dL HbA1c = 6.0 ± 0.81% **NFAI:** FPG = 109.3 ± 60.05 mg/dL HbA1c = 5.8 ± 0.49% **Controls:** FPG = 101.7 ± 22.30 mg/dL HbA1c = 6.1 ± 0.80% **FPG:** ACS vs. NFAI vs. controls, *p* = 0.768 **HbA1c:** ACS vs. NFAI vs. controls, *p* = 0.305
Adamska 2022 [56] Retrospective study	**N = 295 AI** **MACS** = 56 (c-1 mg-DST 1.8–5 µg/dL) F/M = 38/18 Median age = 64 y (range: 58–71) y **NFAI** = 239 (c-1 mg-DST ≤ 1.8 µg/dL) F/M = 158/81 Median age = 62 y (range: 54–68) y	Patients using antidiabetic medication were thought to have T2DM.	**MACS:** T2DM = 41% **NFAI:** T2DM = 23% **T2DM:** MACS vs. NFAI, *p* < 0.01	**MACS:** FPG = 97 (range: 92–114) mg/dL **NFAI:** FPG = 95 (range: 89–105) mg/dL **FPG:** MACS vs. NFAI, *p* = 0.7
Ouyang 2022 [57] Retrospective study	**N = 98 MACS** F/M = 65/33 Mean age = 51.1 ± 10.3 y **Females with MACS** = 65 Mean age = 50.5 ± 9.8 y **Males with MACS** = 33 Mean age = 52.2 ± 11.4 y **MACS:** ■ low-cortisol group c-1 mg-DST 1.8–5 µg/dL ■ high-cortisol group c-1 mg-DST > 5 µg/dL	T2DM diagnosis: a confirmed diagnosis of the disease and at least one glucose-lowering medication.	**MACS (F + M):** T2DM = 24.5% **Females with MACS:** T2DM = 20% **Males with MACS:** T2DM = 33.3% **T2DM:** Females with MACS vs. Males with MACS, *p* = 0.147	**Females with MACS:** FPG = 5.2 ± 1.5 mmol/L FCP = 2.4 (1.9–3.0) mU/L HbA1c = 6.0 ± 1.5% **Males with MACS:** FPG = 5.5 ± 1.4 mmol/L FCP = 3.2 (2.1–3.7) mU/L HbA1c = 6.1 ± 1.0% **FPG:** Females with MACS vs. Males with MACS, *p* = 0.320 **FPG:** Females with MACS vs. Males with MACS, *p* = 0.114 **HbA1c:** Females with MACS vs. Males with MACS, *p* = 0.610
Yano, 2022 [58] Retrospective cross-sectional study	**N = 194 with adrenal tumours** **ACS** = 97 (c-1 mg-DST ≥ 1.8 µg/dL) **ACS was further divided** into overt CS (N = 17) and subclinical CS (N = 80) F/M = 60/37 Median age = 62.0 y (range: 45–69) y **NFAI** = 97 (c-1 mg-DST < 1.8 µg/dL) F/M = 52/45 Median age = 58.0 y (range: 51–67) y	FPG ≥ 126 mg/dL HbA1c ≥ 6.5% Previous therapy for diabetes	**ACS:** T2DM = 44% **NFAI:** T2DM = 22% **T2DM:** ACS vs. NFAI, *p* = 0.002	NA
Araujo-Castro 2021 [59] Retrospective observational study	**N = 642 AI** **ACS** = 337 (c-1 mg-DST > 1.8 µg/dL) F/M = 194/143 Mean age = 65.0 ± 10.6 y **NFAI** = 305 (c-1 mg-DST ≤ 1.8 µg/dL F/M = 169/136 Mean age = 61.5 ± 10.2 y	NA	**ACS:** T2DM = 32.1% **NFAI:** T2DM = 24.3% **T2DM:** ACS vs. NFAI, *p* = 0.031	**ACS:** FPG = 112.3 ± 35.6 mg/dL HbA1c = 6.5 ± 1.4% **NFAI:** FPG = 105.9 ± 5.87 mg/dL HbA1c = 6.6 ± 5.87% **HbA1c:** ACS vs. NFAI, *p* = 0.832 **FPG:** ACS vs. NFAI, *p* = 0.007
Araujo-Castro 2021 [60] Retrospective multi-centre study	**N = 823 AI** F/M = 472/351 Mean age = 63.1 ± 11.0 y **ACS1** = 276 (c-1 mg-DST ≥ 1.8 µg/dL) **ACS2** = 113 (c-1 mg-DST ≥ 3 µg/dL) **ACS3** = 46 (c-1 mg-DST ≥ 5 µg/dL) **NFAI** = 710	FPG ≥ 126 mg/dL 2 h-plasma glucose ≥ 200 mg/dL during OGTT HbA1c ≥ 6.5% In a patient with classic symptoms of hyperglycaemia or hyperglycaemic crisis, a random plasma glucose ≥ 200 mg/dL (ADA 2018)	Prevalence of T2DM among AI was 26.0% Prevalence of T2DM **ACS 1 vs. NFAI** OR = 1.6, 95% CI: 1.2–2.2, *p* = 0.005 **ACS 2 vs. NFAI** OR = 1.7, 95% CI: 1.1–2.6, *p* = 0.014 **ACS 3 vs. NFAI** OR = 1.2, 95% CI: 0.6–2.3, *p* = 0.654	**ACS 1 vs. NFAI** **FPG:** 111.7 ± 36.2 vs. 104.9 ± 2 8.7 mg/dL, *p* = 0.004 **HbA1c:** 6.5 ± 1.2 vs. 6.3 ± 4.4%, *p* = 0.738 **ACS 2 vs. NFAI** **FPG:** 112.5 ± 40.8 vs. 106.3 ± 29.8 mg/dL, *p* = 0.055 **HbA1c:** 6.5 ± 1.2 vs. 6.4 ± 3.9%, *p* = 0.861 **ACS 3 vs. NFAI** **FPG:** 99.1 ± 28.3 vs. 107.6 ± 31.7 mg/dL, *p* = 0.081 **HbA1c:** 6.2 ± 0.8 vs. 6.4 ± 3.7%, *p* = 0.860
Singh 2020 [61] Retrospective study	**N = 443 adrenal adenoma** **MACS** = 168 (c-1 mg-DST 1.9–5 µg/dL) F/M = 113/55 Median age = 66.1 y (range: 29.9–91.2) y **NFAI** = 275 (c-1 mg-DST < 1.9 µg/dL) F/M = 167/108 Median age = 59.5 y (range: 20.8–83.9) y	NA	**MACS:** T2DM = 41.9% **NFAI:** T2DM = 40.1% **T2DM:** MACS vs. NFAI, *p* = 0.801	NA
Falcetta 2020 [62] Retrospective study	**N = 310 AI** F/M = 200/110 Mean age = 58.3 ± 12.9 y **ACS** = 81 (c-1 mg-DST > 5 µg/dL or >1.8 and ≤5 μg/dL and at least one of the following: low ACTH, increased 24-h UFC, absence of cortisol rhythm, and post-LDDST cortisol level > 1.8 μg/dL) F/M = 57/24 Mean age = 62.0 ± 12.8 y **NFAI** = 209 (c-1 mg DST < 1.8 µg/dL) F/M = 132/77 Mean age = 57.3 ± 12.1 y (20 patients with overt adrenal hyperfunction were excluded)	FPG ≥ 126 mg/dL 2 h-plasma glucose ≥ 200 mg/dL during OGTT HbA1c ≥ 6.5% In a patient with classic symptoms of hyperglycaemia or hyperglycaemic crisis, a random plasma glucose ≥ 200 mg/dL (ADA 2020)	**AI:** T2DM = 19.4% **ACS:** T2DM = 17.3% **NFAI:** T2DM = 18.7% **T2DM:** ACS vs. NFAI, *p* = 0.786	**AI:** IFG = 9% IGT = 3.9% **ACS:** IFG = 13.6% IGT = 4.9% **NFAI:** IFG = 8.1% IGT = 2.9% **IFG:** ACS vs. NFAI, *p* = 0.159 **IGT:** ACS vs. NFAI, *p* = 0.473
Di Dalmazi 2020 [63] Retrospective study	**N = 632 AI** **ACS** = 212 (c-1 mg-DST 1.9–5 µg/dL) F/M = 135/77 Median age = 65.8 y (range: 58.1–72.4) y **NFAI** = 420 (c-1 mg-DST < 1.8 µg/dL) F/M = 249/171 Median age = 60.9 y (range: 52.1–68.7) y	FPG ≥ 126 mg/dL 2 h-plasma glucose ≥ 200 mg/dL during OGTT HbA1c ≥ 6.5% In a patient with classic symptoms of hyperglycaemia or hyperglycaemic crisis, a random plasma glucose ≥ 200 mg/dL (ADA 2010)	**ACS:** T2DM = 28.3% **NFAI:** T2DM = 16.0% **T2DM:** NFAI vs. ACS, *p* < 0.001	**ACS:** HbA1c = 49.2 (40.7–58.7) mmol/moL **NFAI:** HbA1c = 49.7 (43.2–61.8) mmol/moL **HbA1c:** ACS vs. NFAI, *p* = 0.815
Podbregar 2020 [64] Retrospective cross-sectional study	**N = 432 AI** F/M = 253/179 Median age = 63.4 y (range: 54.0–71.6) y **ACS** = 142 (c-1 mg-DST ≥ 1.8 µg/dL 128 out of 142 patients had serum cortisol after 1 mg DST between 1.8–5 µg/dL and 14 patients had cortisol levels > 5 µg/dL Mean age = 64.90 ± 12.08 y **NFAI** = 290 (c-1 mg-DST < 1.8 µg/dL) Mean age = 61.76 ± 11.13 y	NA	**AI:** T2DM = 12% **T2DM:** ACS vs. NFAI, *p* > 0.05	**ACS:** FPG = 5.78 ± 1.71 mmol/L **NFAI:** FPG = 5.75 ± 1.40 mmol/L **FPG:** ACS vs. NFAI, *p* = 0.876
Reimondo 2020 [65] Prospective cohort study	**N = 601 individuals who have had computed tomography scans** F/M = 270/331 Mean age = 63.5 ± 14.4 y **AI** = 44 F/M = 12/32 Mean age = 65.6 ± 10.3 y **ACS** = 20 patients with c-1 mg-DST ≥ 1.8 µg/dL F/M = 6/14 Mean age = 67.5 ± 9.5 y **NFAI** = 20 patients with c-1 mg-DST < 1.8 µg/dL F/M = 4/16 Mean age = 62.6 ± 8.2 y **Controls** = 557 (patients without AI) F/M = 258/299 Mean age = 63.3 ± 14.7 y	T2DM was defined when the patient reported: FPG ≥ 126 mg/dL 2 h-OGTT ≥ 200 mg/dL HbA1c ≥ 6.5% Previous diagnosis of diabetes, or use of antidiabetic medications	**AI:** T2DM = 31.8% **ACS:** T2DM = 35% **NFAI:** T2DM = 30% **Controls:** T2DM = 14.2% **T2DM:** AI vs. controls, *p* = 0.004 ACS vs. NFAI, *p* = 0.74	**AI:** HbA1c = 7.8 ± 2.9% **ACS:** HbA1c = 8.9 ± 3.8% **NFAI:** HbA1c = 7.1 ± 1% **Controls:** HbA1c = 7.2 ± 2.6% **HbA1c:** AI vs. controls, *p* = 0.14 ACS vs. NFAI, *p* = 0.11
Ueland 2020 [66] Retrospective study	**N = 165 AI** **ACS** = 83 (c-1 mg-DST > 1.8 µg/dL) F/M = 58/25 Median age = 65 y (range: 29–86) y **NFAI** = 82 F/M = 48/34 Median age = 68.5 y (range: 33–82) y	NA	**ACS:** T2DM = 16.7% **NFAI:** T2DM = 8.5%	**ACS:** HbA1c = 5.8 (range: 5–9)% **NFAI:** HbA1c = 5.7 (range: 5–9)%
Moraes 2020 [67] Cross-sectional study	**N = 75 AI** **ACS** = 30 (c-1 mg-DST 1.9–5 µg/dL) F/M = 26/4 Median age = 60 y (range: 42–77) y **NFAI** = 45 (c-1 mg-DST ≤ 1.8 µg/dL) F/M = 32/13 Median age = 59 y (range: 32–76) y	NA	**ACS:** T2DM = 40% **NFAI:** T2DM = 31.1% **T2DM:** ACS vs. NFAI, *p* = 0.43	**ACS:** HbA1c = 6.1 (range: 4.6–11.1)% **NFAI:** HbA1c = 6.0 (range: 4.9–7.7)% **HbA1c:** ACS vs. NFAI, *p* = 0.57

Abbreviations: ADA = American Diabetes Association; ACS = autonomous cortisol secretion; ACTH = adrenocorticotropic hormone; AI = adrenal incidentaloma; c-1 mg-DST = serum cortisol after 1 mg dexamethasone suppression test; CS = Cushing’s syndrome; F = female; FCP = fasting C-peptide; FPG = fasting plasma glucose; HbA1c = glycated haemoglobin; IFG = impaired fasting glucose; IGT = impaired glucose tolerance; LDDST = cortisol after 2 days of a low-dose (2 mg/day) dexamethasone suppression test; M = male; MACS = mild autonomous cortisol secretion; N = number of patients; NA = not available; NFAI = non-functioning adrenal incidentaloma; OGTT = oral glucose tolerance test; OR = odds ratios; T2DM = type 2 diabetes mellitus; SD = standard deviation; UFC = urinary free cortisol; vs. = versus; y = years.

**Table 2 biomedicines-12-01606-t002:** Summary of the studies analysing the glucose profile in NFAI compared to controls (without any adrenal adenomas); the display starts with the most recent publication date [68,69,70,71].

First Author Year of Publication Reference Study Design	Studied Population Number of Patients Age (Years) Gender (F/M)	Criteria for the Diagnosis of T2DM	The Prevalence of T2DM	Glucose Parameters [(Mean ± SD or Median (IQR)]
Yilmaz 2022 [68] Retrospective study	**N = 85 patients with or without adrenal adenomas** **NFAI** = 43 (c-1 mg-DST < 1.8 µg/dL) F/M = 24/19 Mean age = 64.6 ± 11.5 y **Controls** = 42 (healthy individuals matched to the NFAI group in terms of age, gender, BMI, diabetes) F/M = 23/19 Mean age = 64.1 ± 11.8 y	NA	**NFAI:** T2DM = 27.9% **Controls:** T2DM = 26.1% **T2DM:** NFAI vs. controls, *p* = 0.209 **T2DM** was significantly correlated with masked hypertension, OR = 2.07, *p* = 0.044	**FPG** **NFAI:** 106.8 ± 15.3 mg/dL **Controls:** 107.3 ± 14.9 mg/dL FPG: NFAI vs. controls, *p* = 0.128
Zhang 2021 [69] Cohort study	**N = 2008 patients with or without adrenal adenomas** **N1 = 1004 with non-secreting adrenal tumours** F/M = 582/422 Median age = 63 y (range: 21–96) y **NFAI** = 141 (c-1 mg-DST ≤ 1.8 µg/dL **MACS** = 81 (c-1 mg-DST > 1.8 µg/d) [No DST performed = 782 (unknown cortisol secretion)] **Controls** = 1004 (age- and sex-matched referent subjects without adrenal tumour) F/M = 582/422 Median age = 63 y (range: 21–96) y	T2DM was determined by HbA1c ≥ 6.5% Dysglycemia: composite of prediabetes or diabetes mellitus.	**NFAI:** T2DM = 27.5% Dysglycemia = 43.1% **Controls:** T2DM = 17.4% Dysglycemia = 28% **T2DM:** NFAI vs. controls, *p* < 0.001 **Dysglycemia:** NFAI vs. controls, *p* < 0.001	NA
Kim 2020 [70] Cross-sectional study	**N = 616 patients with or without adrenal adenomas** **NFAI** = 154 (c-1 mg-DST ≤ 1.8 µg/dL) F/M = 40/114 Mean age = 55.7 ± 8.6 y **Controls** = 462 (age and sex-matched control group without adrenal tumours) F/M = 126/336 Mean age = 55.7 ± 8.9 y	HbA1c ≥ 6.5% Previous therapy for diabetes	**NFAI:** T2DM = 25.3% **Controls:** T2DM = 14.5% **T2DM:** NFAI vs. controls, *p* = 0.003 OR = 1.89, 95% CI: 1.17–3.06	**NFAI:** FPG = 108.0 ± 26.5 mg/dL HbA1c = 6.1 ± 0.9% **Controls:** FPG = 99.5 ± 17.7 mg/dL HbA1c = 5.9 ± 0.6% **FPG:** NFAI vs. controls, *p* < 0.001 **HbA1c:** NFAI vs. controls *p* = 0.009
Paula 2020 [71] Cross-sectional study	**N = 82 patients with or without adrenal adenomas** **NFAI**^#^ = 42 (c-1 mg-DST < 1.8 µg/dL) F/M = 8/34 Mean age = 58.4 ± 8.57 y **Controls** ^#^ = 40 (without adrenal tumours) F/M = 9/31 Mean age = 58.1 ± 10.65 y	FPG ≥ 126 mg/dL 2 h-plasma glucose ≥ 200 mg/dL during OGTT HbA1c ≥ 6.5% In a patient with classic symptoms of hyperglycaemia or hyperglycaemic crisis, a random plasma glucose ≥ 200 mg/dL (ADA 2017)	**NFAI:** T2DM = 45.2% **Controls:** T2DM = 35% **T2DM:** NFAI vs. controls, *p* = 0.38	**NFAI:** FPG = 105 (range: 71–217) mg/dL HbA1c = 5.9 (range: 4.3–10.8)% **Controls:** FPG = 97.5 (range: 71–152) mg/dL HbA1c = 5.7 (range: 4.1–8.1)% **FPG:** NFAI vs. controls, *p* = 0.18 **HbA1c:** NFAI vs. controls, *p* = 0.94

Abbreviations: ADA = American Diabetes Association; BMI = body mass index; c-1 mg-DST = serum cortisol after 1 mg dexamethasone suppression test; F = female; FPG = fasting plasma glucose; HbA1c = glycated haemoglobin; M = male; MACS = mild autonomous cortisol secretion; N = number of patients; NA = not available; NFAI = non-functioning adrenal incidentaloma; OGTT = oral glucose tolerance test; OR = odds ratios; T2DM = type 2 diabetes mellitus; vs. = versus; y = years; ^#^ atherosclerotic cardiovascular disease, statin use, smoking status, age, gender, and ethnicity did not differ between the groups.

**Table 3 biomedicines-12-01606-t003:** Summary of the studies analysing the glucose profile in subjects confirmed with NFAI compared to the group of patients diagnosed with possible ACS or ACS (the display starts with the most recent publication date [74,75,76,77,78].

First Author Year of Publication Reference Study Design	Studied Population Number of Patients Age (Years) Gender (F/M)	Criteria for the Diagnosis of T2DM	The Prevalence of T2DM	Glucose Parameters [(Mean ± SD or Median (IQR)]
Deutschbein, 2022 [74] Retrospective multicentre cohort study	**N = 3656 patients with AI** **ACS** = 247 (c-1 mgDST > 5 µg/dL) F/M = 169/78 Median age = 63 y (range: 55–70) y **possible ACS** = 1320 (c-1 mg-DST 1.8–5 µg/dL) F/M = 860/460 Median age = 63 y (range: 56–70) y **NFAI** = 2089 NFAI (c-1 mg-DST < 1.8 µg/dL) F/M = 1321/768 Median age = 60 y (range: 52–67) y	NA	**AI:** T2DM = 20.5% **ACS:** T2DM = 26.7% **Possible ACS:** T2DM = 23% **NFAI:** T2DM = 18.2% **T2DM:** ASC vs. possible ACS vs. NFAI, *p* < 0.001	NA
Prete 2022 [76] Cross-sectional study	**N = 1305 patients with benign adrenocortical adenomas** F/M = 878/427 Median age = 60 y (interquartile range: 52–67) y **Adrenal CS** = 65 F/M = 56/9 Median age = 48 y (interquartile range: 38–60) y **MACS** = 140 (c-1 mgDST > 5 µg/dL) F/M = 103/37 Median age = 63 y (interquartile range: 54–69) y **possible MACS** = 451 (c-1 mg-DST 1.8–5 µg/dL) F/M = 303/148 Median age = 64 y (interquartile range: 56–71) y **NFAI** = 649 (c-1 mg-DST < 1.8 µg/dL) F/M = 416/233 Median age = 58 y (interquartile range: 51–65) y	FPG ≥ 126 mg/dL 2 h-plasma glucose ≥ 200 mg/dL during OGTT HbA1c ≥ 6.5% In a patient with classic symptoms of hyperglycaemia or hyperglycaemic crisis, a random plasma glucose ≥ 200 mg/dL **Prediabetes:** HbA1c of 5.7% to 6.4% (ADA 2021)	**Adrenal CS:** T2DM = 31.5% **MACS:** T2DM = 33.7% **Possible MACS:** T2DM = 32.2% **NFAI:** T2DM = 26.4%	NA
Sojat 2021 [75] Case-control study	**N = 92 patients with or without adrenal adenomas** **Possible ACS** = 34 (c-1 mg-DST 1.8–5 µg/dL) F/M = 29/5 Mean age = 56.65 ± 5.61 y **NFAI** = 26 (c-1 mg-DST < 1.8 µg/dL) F/M = 18/8 Mean age = 58.54 ± 8.81 y **Healthy controls** = 32 (without adrenal tumours) F/M = 25/7 Mean age = 57.59 ± 9.36 y	FPG ≥ 126 mg/dL 2 h-plasma glucose ≥ 200 mg/dL during OGTT HbA1c ≥ 6.5% In a patient with classic symptoms of hyperglycaemia or hyperglycaemic crisis, a random plasma glucose ≥ 200 mg/dL (ADA 2010)	**Possible ACS:** T2DM = 20.6% **NFAI:** T2DM = 15.4% **Healthy controls:** T2DM = NA **T2DM:** NFAI vs. possible ACS, *p* = 0.606	**Possible ACS:** HbA1c = 5.82 ± 0.57% FPG = 5.55 ± 0.67 mmol/L **NFAI:** HbA1c = 5.76 ± 0.83% FPG = 5.50 ± 0.75 mmol/L **Healthy controls:** HbA1c = NA FPG = 4.93 ± 0.39 mmol/L **HbA1c:** NFAI vs. possible ACS, *p* = 0.802 **FPG:** NFAI vs. possible ACS, *p* = 0.003
Yilmaz 2021 [78] Retrospective study	**N = 755 patients with AI** F/M = 497/258 Median age = 56 y (range: 18–86) y **ACS** = 37 (c-1 mg DST > 5 µg/dL) **possible ACS** = 82 (c-1 mg-DST 1.9–5 µg/dL) **NFAI** = 542 (c-1 mg-DST ≤ 1.8 µg/dL) **functional adenomas** (CS, Pheochromocytoma, hyperaldosteronism) = 94	NA	**AI:** T2DM = 28.3%	NA
Naka 2020 [77] Cross-sectional study	**N = 339 patients with adrenal tumours** **CS** = 23 (c-1 mg-DST > 5 µg/dL) F/M = 19/4 Mean age = 49.8 ± 3.2 y **possible ACS** = 84 (c-1 mg-DST 1.8–5 µg/dL) F/M = 42/42 Mean age = 64.1 ± 1.2 y **NFAI** = 232 (c-1 mg-DST < 1.8 µg/dL) F/M = 122/110 Mean age = 58.9 ± 0.8 y	FPG ≥ 126 mg/dL Casual plasma glucose ≥ 200 mg/dL 2 h-plasma glucose ≥ 200 mg/dL during OGTT Previous therapy for diabetes (ADA 2004)	**CS:** T2DM = 13% **Possible ACS:** T2DM = 33.3% **NFAI:** T2DM = 24.2% **T2DM:** CS vs. possible ACS vs. NFAI, *p* = 0.09	NA

**Abbreviations**: ADA = American Diabetes Association; ACS = autonomous cortisol secretion; AI = adrenal incidentaloma; c-1 mg-DST = serum cortisol after 1 mg dexamethasone suppression test; CS = Cushing’s syndrome; F = female; FPG = fasting plasma glucose; HbA1c = glycated haemoglobin; M = male; MACS = mild autonomous cortisol secretion; N = number of patients; NA = not available; NFAI = non-functioning adrenal incidentaloma; OGTT = oral glucose tolerance test; T2DM = type 2 diabetes mellitus; vs. = versus; y = years.

**Table 4 biomedicines-12-01606-t004:** Summary of the studies analysing the glucose profile in subjects confirmed with NFAI compared to a group of patients diagnosed with Cushing syndrome (the display starts with the most recent publication date) [82,83].

First Author Year of Publication Reference Study Design	Studied Population Number of Patients Age (Years) Gender (F/M)	Criteria for the Diagnosis of T2DM	The Prevalence of T2DM	Glucose Parameters
Delivanis 2022 [82] Cross-sectional study	**N = 227 patients with adrenal adenomas** CS = 20 F/M = 18/2 Median age = 46 y (range: 18–69) y **MACS** = 76 (c-1 mg or 8 mg-DST of >1.8 µg/dL in a patient without features of overt CS) F/M = 42/34 Median age = 58 y (range: 28–87) y **NFAI** = 131 (c-1 mg DST ≤ 1.8 µg/dL) F/M = 91/40 Median age = 57 y (range: 18–89) y	T2DM or prediabetes diagnosis: treatment included at least one glucose-lowering drug.	**CS:** T2DM or IFG = 35% **MACS:** T2DM or IFG = 45% **NFAI:** T2DM or IFG = 34% **T2DM or IFG:** CS vs. MACS vs. NFAI, *p* = 0.271	NA
Athimulam 2020 [83] Cross-sectional study	**N = 213 patients with AI** **CS** = 22 F/M = 18/4 Median age = 41.5 y (range: 18–61) y **MACS** = 92 (c-1 mg-DST > 1.8 µg/dL) F/M = 57/35 Median age = 59.5 y (range: 28–82) y **NFAI** = 99 (c-1 mg-DST ≤ 1.8 µg/dL) F/M = 67/32 Median age = 59 y (range: 28–93) y	HbA1c ≥ 6.4% Previous therapy for diabetes	**CS:** T2DM = 41% **MACS:** T2DM = 41% **NFAI:** T2DM = 41% **T2DM:** CS vs. MACS vs. NFAI, *p* = 0.99 MACS vs. NFAI, *p* = 0.98	NA

**Abbreviations:** ACS = autonomous cortisol secretion; AI = adrenal incidentaloma; c-1 mg-DST = serum cortisol after 1 mg dexamethasone suppression test; CS = Cushing’s syndrome; F = female; HbA1c = glycated haemoglobin; IFG = impaired fasting glucose; M = male; MACS = mild autonomous cortisol secretion; N = number of patients; NA = not available; NFAI = non-functioning adrenal incidentaloma; T2DM = type 2 diabetes mellitus; vs. = versus; y = years.

**Table 5 biomedicines-12-01606-t005:** Summary of the studies analysing the glucose profile in terms of pre-diabetes (IFG and/or IGT) in subjects confirmed with NFAI (the display starts with the most recent publication date) [56,69,70,85].

First Author Reference Year of Publication Study Design	Studied Population Number of Patients Age (Years) Gender (F/M)	Criteria for the Diagnosis of Prediabetes	The Prevalence of Prediabetes Additional Glucose Profile Data (Mean ± Standard Deviation or Median (Range)
Szychlinska 2023 [85] Case-control study	**N = 92 patients with or without adrenal adenomas** **NFAI** = 48 (c-1 mg-DST < 1.8 µg/dL) F/M = 32/16 Mean age = 58.6 ± 9 y **Controls** = 44 (matched for age, gender and BMI) F/M = 29/15 Mean age = 57 ± 7 y (patients with T2DM were excluded)	**IFG** was defined as FPG levels between 100 and 125 mg/dL **IGT** was defined as 2-h plasma glucose during 75-g OGTT levels between 140 and 199 mg/dL (ADA 2020)	**NFAI:** FPG = 96.1 ± 12 mg/dL IFG = 20.8% Fasting insulin = 11.4 ± 4.9 uU/mL IGT = 27% 2 h-OGTT = 127.7 ± 38.2 mg/dL HOMA-IR = 2.72 ± 1.23 **Controls** FPG = 100.4 ± 11 mg/dL Fasting insulin = 8.9 ± 5.8 uU/mL IFG = 47.7% IGT = 0.9% 2 h-OGTT = 105.1 ± 27.9 mg/dL HOMA-IR = 2.26 ± 1.64 **FPG:** NFAI vs. controls, *p* = 0.7 **Fasting insulin:** NFAI vs. controls, *p* = 0.03 **IFG:** NFAI vs. controls, *p* < 0.01 **IGT:** NFAI vs. controls, *p* = 0.026 **2 h-OGTT:** NFAI vs. controls, *p* = 0.04 **HOMA-IR:** NFAI vs. controls *p* = 0.13
Adamska 2022 [56] Retrospective study	**N = 295 patients with AI** **MACS** = 56 (c-1 mg-DST 1.8–5 µg/dL) F/M = 38/18 Median age = 64 y (range: 58–71) y **NFAI** = 239 (c-1 mg-DST ≤ 1.8 µg/dL) F/M = 158/81 Median age = 62 y (range: 54–68) y	Prediabetes state was defined as impaired **FPG** = 100–125 mg/dL and/or **IGT** (serum glucose level of 140–199 mg/dL in the 2 h-OGTT	**MACS:** Prediabetes = 26.8% **NFAI:** Prediabetes = 34.3% **Prediabetes:** MACS vs. NFAI, *p* = 0.35 **MACS:** FPG = 97 (range: 92–114) mg/dL **NFAI** FPG = 95 (range: 89–105) mg/dL **FPG:** MACS vs. NFAI, *p* = 0.7
Zhang 2021 [69] Cohort study	**N = 2008 patients with or without adrenal adenomas** **N = 1004 with non-secreting adrenal tumours** F/M = 582/422 Median age = 63 y (range: 21–96) y **NFAI** = 141 (c-1 mg-DST ≤ 1.8 µg/dL **MACS** = 81 (c-1 mg-DST > 1.8 µg/dL) **Controls** = 1004 (age- and sex-matched referent subjects without adrenal tumour) F/M = 582/422 Median age = 63 y (range: 21–96) y	Prediabetes was defined by HbA1c between 5.7% and 6.4%.	**NFAI:** Prediabetes = 15.4% **Controls:** Prediabetes = 10.5% **Prediabetes:** NFAI vs. controls, *p* < 0.001
Kim 2020 [70] Cross-sectional study	**N = 616 patients with or without adrenal adenomas** **NFAI** = 154 (c-1 mg-DST ≤ 1.8 µg/dL) F/M = 40/114 Mean age = 55.7 ± 8.6 y **Controls** = 462 (age and sex-matched control group without adrenal tumours) F/M = 126/336 Mean age = 55.7 ± 8.9 y	NA	**NFAI:** HOMA-IR = 2.80 ± 2.17 **Controls:** HOMA-IR = 2.00 ± 1.10 **HOMA-IR:** NFAI vs. controls, *p* = 0.022

**Abbreviations:** ADA = American Diabetes Association; ACS = autonomous cortisol secretion; AI = adrenal incidentaloma; BMI = body mass index; c-1 mg-DST = serum cortisol after 1 mg dexamethasone suppression test; F = female; FPG = fasting plasma glucose; HbA1c = glycated haemoglobin; HOMA-IR = homeostasis model assessment of insulin resistance; IFG = impaired fasting glucose; IGT = impaired glucose tolerance; M = male; MACS = mild autonomous cortisol secretion; N = number of patients; NA = not available; NFAI = non-functioning adrenal incidentaloma; OGTT = Oral glucose tolerance test; T2DM = type 2 diabetes mellitus; vs. = versus; y = years.

**Table 6 biomedicines-12-01606-t006:** Summary of the longitudinal studies analysing the glucose profile in NFAI and/or MACS (the display starts with the most recent publication date) [52,54,59,62,69,70,87,88].

First Author Year of Publication Reference Study Design Follow-Up Period of Time	Study Population Number of Patients Age (Years) Gender (F/M)	Glucose Profile at Baseline	Glucose Profile According to the Follow-Up
Candemir 2024 [88] Retrospective study follow-up: 2 y	**N = 207 patients with AI** **NFAI** = 80 (c-1 mg-DST ≤ 1.8 µg/dL) F/M = 53/27 Mean age = 60 ± 12 y **Controls** = 127 (without any adrenal pathology, matched for age, sex, and BMI, FPG, HbA1c) F/M = 91/36 Mean age = 59 ± 13 y	**NFAI:** FPG = 93.07 ± 9.7 8 mg/dL HbA1c = 4.86 ± 0.35% **Controls:** FPG = 90.81 ± 6.83 mg/dL HbA1c = 4.79 ± 0.23% **FPG:** NFAI vs. controls, *p* = 0.073 **HbA1c:** NFAI vs. controls, *p* = 0.079	After 2 years, **FPG** levels increased in both groups (*p* < 0.001). **Prediabetes** was developed by 17.5% (N = 14) of the patients in the adenoma group compared to 1.6% (N = 2) of the individuals in the control group (*p* < 0.001).
Favero 2024 [52] Cross-sectional study Follow-up: 24.9 ± 5.3 months	**N = 126 patients with AI** (longitudinal arm) Mean age = 63.5 ± 9.5 y (range: 27–83) y F/M = 80/46 **MACS** = 66 (c-1 mg-DST > 1.8 µg/dL) F/M = 45/21 Mean age = 65.5 ± 8.0 y (range: 40–83) y **NFAI** = 60 (c-1 mg-DST ≤ 1.8 µg/dL µg/dL) F/M = 35/25 Mean age = 61.5 ± 10.6 y (range: 27–80) y	**AI:** T2DM = 16.4% **MACS:** T2DM = 17.8% **NFAI:** T2DM = 15% **T2DM:** MACS vs. NFAI, *p* = 0.246	**AI:** T2DM = 25.4% **MACS:** T2DM = 15.9% **NFAI:** T2DM = 20% **T2DM:** MACS vs. NFAI, *p* = 0.221
Brox-Torrecilla 2023 [54] Retrospective study Follow-up: 28 months	**N = 709 patients with AI** F/M = 397/312 Mean age = 63.4 ± 10.8 y **ACS** = 231 (c-1 mg-DST ≥ 1.8 µg/dL) F/M = 131/100 Mean age = 65.0 ± 10.60 y **NFAI** = 478 (c-1 mg-DST < 1.8 µg/dL) F/M = 267/211 Mean age = 62.7 ± 10.77 y	**ACS:** T2DM = 27.7% T2DM and HbA1c > 7% = 35.4% T2DM and HbA1c > 8% = 14.6% **NFAI:** T2DM = 22.6% T2DM and HbA1c > 7% = 37.8% T2DM and HbA1c > 8% = 9.5% **T2DM**: ACS vs. NFAI, *p* = 0.137 **T2DM and HbA1c > 7%:** ACS vs. NFAI, *p* = 0.787 **T2DM and HbA1c > 8%:** ACS vs. NFAI, *p* = 0.386	A new diagnosis of T2DM had been identified for 24 individuals; there were no group differences in the incidence of T2DM (HR = 1.17, 95% CI: 0.52–2.64)
Araujo-Castro 2021 [59] Retrospective observational study Follow-up: mean 41.3 months	**N = 642 patients with AI** **ACS** = 337 (c-1 mg-DST > 1.8 µg/dL) F/M = 194/143 Mean age = 65.0 ± 10.6 y **NFAI** = 305 (c-1 mg DST ≤ 1.8 µg/dL) F/M = 169/136 Mean age = 61.5 ± 10.2 y	**ACS**: T2DM = 32.1% **NFAI**: T2DM = 24.3% **T2DM**: ACS vs. NFAI, *p* = 0.031	NFAI (N = 273), development of **T2DM** during follow-up = 5.2% **NFAI progressing to ACS** (N = 32), development of T2DM during follow-up = 9.5% HR = 1.65, 95% CI: 0.36–7.66, *p* = 0.543 Incident **T2DM** in 5.7% during follow-up
Zhang 2021 [69] Cohort study Follow-up: 7.2 y (NFAI) and 6.8 y (controls)	**N = 2008 patients with or without adrenal adenomas** **N = 1004 with non-secreting adrenal tumours** F/M = 582/422 Median age = 63 y (range: 21–96) y **NFAI** = 141 (c-1 mg-DST ≤ 1.8 µg/dL **MACS** = 81 (c-1 mg-DST > 1.8 µg/dL) **Controls** = 1004 (age- and sex-matched referent subjects without adrenal tumour) F/M = 582/422 Median age = 63 y (range: 21–96) y	**NFAI:** Prediabetes = 15.4% T2DM = 27.5% Dysglycemia = 43.1% **Controls:** Prediabetes = 10.5% T2DM = 17.4% Dysglycemia = 28% **Prediabetes:** NFAI vs. controls, *p* < 0.001 **T2DM**: NFAI vs. controls, *p* < 0.001 **Dysglycemia:** NFAI vs. controls, *p* < 0.001	Patients with adrenal adenomas had a higher unadjusted 10-year cumulative incidence of dysglycemia than the control group (18% vs. 14%) During the follow-up period, patients with MACS compared to NFAT had higher unadjusted overall mortality (3% vs. 2% at 5 years, 20% vs. 9% at 10 years, and 37% vs. 19% at 15 years)
Podbregar, 2021 [87] Prospective study Follow-up: 10.5 y	**N = 67 patients with NFAI** (c-1 mg-DST ≤ 1.8 µg/dL) F/M = 47/20 Mean age = 57.9 y	T2DM = 0.03%	T2DM = 17.9% T2DM was 0.03% at baseline and 17.9% at the follow-up (*p* = 0.002)
Falcetta 2020 [62] Retrospective study Follow-up: 31.4 months	**N = 310 patients with AI** F/M = 200/110 Mean age = 58.3 ± 12.9 y **ACS** = 81 (c-1 mg-DST > 5 µg/dL or >1.8 and ≤5 μg/dL and at least one of the following: low ACTH, increased 24-h UFC, absence of cortisol rhythm, and post-LDDST cortisol level > 1.8 μg/dL) F/M = 57/24 Mean age = 62.0 ± 12.8 y **NFAI** = 209 (c-1 mg DST < 1.8 µg/dL) F/M = 132/77 Mean age = 57.3 ± 12.1 y (20 patients with overt adrenal hyper function were excluded)	**AI:** T2DM = 19.4% IFG = 9% IGT = 3.9% **ACS:** T2DM = 17.3% IFG = 13.6% IGT = 4.9% **NFAI** T2DM = 18.7% IFG = 8.1% IGT = 2.9% **T2DM:** ACS vs. NFAI, *p* = 0.786 **IFG:** ACS vs. NFAI, *p* = 0.159 **IGT:** ACS vs. NFAI, *p* = 0.473	**AI without enlargement (N = 257):** T2DM = 20.2% IFG = 8.2% IGT = 3.5% **AI with enlargement (N = 53):** T2DM = 15.1%, *p* = 0.389 IFG = 13.2%, *p* = 0.289 IGT = 5.7%, *p* = 0.438 **Not developing ACS during follow-up (N = 202)** T2DM = 19.3% IFG = 6.9% IGT = 2.5% **Developing ACS during follow-up (N = 7)** T2DM = 0%, *p* = 0.205 IFG = 42.9%, *p* = 0.001 IGT = 14.3%, *p* = 0.071:
Kim 2020 [70] Cross-sectional study Follow-up: mean 7.5 y	**N = 616 patients with or without adrenal adenomas** **NFAI** = 154 (c-1 mg-DST ≤ 1.8 µg/dL) F/M = 40/114 Mean age = 55.7 ± 8.6 y **Controls** = 462 (age and sex-matched control group without adrenal tumours) F/M = 126/336 Mean age = 55.7 ± 8.9 y	**NFAI:** HbA1c = 6.1 ± 0.9% T2DM = 25.3% **Controls:** HbA1c = 5.9 ± 0.6% T2DM = 14.5% **HbA1c:** NFAI vs. controls *p* = 0.009 **T2DM:** NFAI vs. controls, *p* = 0.003 OR = 1.89, 95% CI; 1.17–3.06	There was no difference in the incidence of T2DM between the NFAI and control groups Adrenal lesions were greater in NFAI participants with diabetes than in those without (*p* = 0.048) at follow-up

**Abbreviations:** ACS = autonomous cortisol secretion; ACTH = adrenocorticotropic hormone; AI = adrenal incidentaloma; CI = confidence intervals; c-1 mg-DST = serum cortisol after 1 mg dexamethasone suppression test; F = female; FPG = fasting plasma glucose; HbA1c = glycated haemoglobin; HR = hazard ratio; IFG = impaired fasting glucose; IGT = impaired glucose tolerance; LDDST = cortisol after 2 days of a low-dose (2 mg/day) dexamethasone suppression; M = male; MACS = mild autonomous cortisol secretion; N = number of patients; NFAI = non-functioning adrenal incidentaloma; OR = odds ratio; T2DM = type 2 diabetes mellitus; UFC = urinary free cortisol; vs. = versus; y = years.

**Table 7 biomedicines-12-01606-t007:** Summary of the longitudinal studies analysing the glucose profile in patients with MACS that underwent adrenalectomy versus conservative management of the adrenal tumours (the display starts with the most recent publication date) [89,90,91,92,93,94].

First Author Publication Year Reference Number Study Design	Studied Population Number of Patients Age (Years) Gender (F/M)	Evolution of Gluco-Metabolic Profile after Adrenalectomy or Conservative Treatment
Remde 2023 [92] Cohort study	**N = 260 patients with AI** F/M = 147/113 Median age = 59.5 y ACS = 41 (c-1 mg-DST > 5 µg/dL) F/M = 25/16 Median age = 56 y **possible ACS** = 96 (c-1 mg-DST 1.9–5 µg/dL) F/M = 53/43 Median age = 63 y **NFAI** = 123 (c-1 mg-DST ≤ 1.8 µg/dL) F/M = 69/54 Median age = 57 y **Conservative vs. Adrenalectomy** **ACS:** 82.1% vs. 17.9% **Possible ACS:** 76.0% vs. 24.0% **NFAI:** 61.0% vs. 39.0% Median follow-up 8.8 years	**Adrenalectomy group:** before vs. after adrenalectomy **T2DM:** 18% vs. 26.2%, *p* = NS **Conservative group:** baseline vs. follow-up **T2DM:** 21.1% vs. 30.2%, *p* = NS **Conservative vs. adrenalectomy** baseline, *p* = NS follow-up, *p* = NS T2DM was considerably less common in non-operated patients with NFAI as compared to possible ACS and ACS at the last follow-up (23.8% vs. 35.6% and 40.0%, *p* < 0.01)
Morelli 2022 [89] Prospective randomized study	**N = 55 patients with AI (c-1 mg-DST 1.8–5 µg/dL)** **Adrenalectomy group** = 25 F/M = 17/8 Mean age = 62.5 ± 10.4 y (range: 41–75) y **Conservative group** = 30 F/M = 24/6 Mean age = 66.1 ± 9.1 y (range: 41–75) y	Adrenalectomy vs. conservative treatmentImproved in glucometabolic control: 28% vs. 3.3%, *p* = 0.02 Worsened in glucometabolic control: 8% vs. 20%, *p* = 0.12 **Baseline vs. 6-month follow-up:** **Adrenalectomy group** **T2DM:** 20% vs. 20% **IGT/IFG:** 28% vs. 20% **HbA1c:** 40.8 ± 6.6 mmol/moL vs. 39.6 ± 5.4 mmol/moL **Conservative group:** T2DM: 20% vs. 20% IGT/IFG: 30% vs. 33.3% HbA1c: 39.5 ± 7.1 mmol/moL vs. 39.8 ± 6.6 mmol/moL
Wang 2022 [93] Retrospective cohort study	**N = 171 patients with AI + surgical approach** F/M = 84/87 Mean age = 50.6 ± 11.4 y (range: 14–78) y **N1** = 41 persistent hypertension **N2** = 130 hypertension resolution AIs with normal hormone levels were enrolled in the study.	The prevalence of T2DM among AI patients was 10.5% **N1 vs. N2** T2DM: 17.1% vs. 8.5%, *p* = 0.202
Thompson 2021 [91] Retrospective study	**N = 271 patients with adrenal tumours undergoing adrenalectomy** **CS** = 127 F/M = 104/23 Mean age = 56.9 ± 12.6 y **ACS** = 45 F/M = 31/14 Mean age = 65.0 ± 10.4 y **NFAI** = 99 F/M = 59/40 Mean age = 60.5 ± 12.1 y **CS and ACS** were diagnosed based on applicable criteria at the time of diagnosis c-1-mg DST for initial evaluation	At the time of surgery **T2DM** CS vs. ACS vs. NFAI: 18.9% vs. 13.3% vs. 14.1% During follow-up, after adrenalectomy patients with ACS showed a slight decrease over time (*p* = NS) in their medication levels with antidiabetics.
Petramala 2020 [94] Cross-sectional study	**N = 628 patients with AI** SH = 157 (c-1 mg-DST > 1.8 µg/dL plus one abnormal: UFC level > 100 µg/24 h, morning plasma ACTH levels < 10 pg/mL) F/M = 64/93 Mean age = 62.9 ± 11 y **NFAI** = 471 (c-1 mg-DST < 1.8 µg/dL) F/M = 189/282 Mean age = 59.6 ± 12.5 y **SH with adrenalectomy** = 29 **SH with conservative treatment** = 118	**Baseline SH vs. NFAI:** T2DM: 19% vs.7%, *p* < 0.05 FPG: 98.9 ± 26.3 mg/dL vs. 99.2 ± 22.3 mg/dL, *p* = NS **Adrenalectomy group** before adrenalectomy vs. after adrenalectomy T2DM 20.7% vs. 7.4%, *p* < 0.05 **Conservative group** baseline vs. follow-up T2DM 26.7% vs. 23.5%, *p* = NS
Sato 2020 [90] Retrospective study	**N = 135 patients with subclinical CS** **Adrenalectomy** = 117 F/M = 76/41 Mean age = 57.6 ± 10.4 y **Conservative group** = 18 F/M = 13/5 Mean age = 65.7 ± 6.8 y	**Conservative vs. Adrenalectomy** T2DM: 55.6% vs. 27%, *p* = 0.0255 **Before vs. after adrenalectomy** Patients with T2DM: HbA1c = 6.83 ± 0.94% vs. HbA1c = 6.09 ± 0.72%, *p* = 0.0019

**Abbreviations:** ACS = autonomous cortisol secretion; ACTH = adrenocorticotropic hormone; AI = adrenal incidentaloma; CS = Cushing’s syndrome; c-1 mg-DST = serum cortisol after 1 mg dexamethasone suppression test; HbA1c = glycated haemoglobin; IFG = impaired fasting glucose; IGT = impaired glucose tolerance; MACS = mild autonomous cortisol secretion; N = number of patients; NFAI = non-functioning adrenal incidentaloma; NS = not statistically significant; SH = subclinical endogenous cortisol excess; T2DM = type 2 diabetes mellitus; vs. = versus; y = years.

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
