# Peer review of "Diabetes Mellitus in Non-Functioning Adrenal Incidentalomas: Analysis of the Mild Autonomous Cortisol Secretion (MACS) Impact on Glucose Profile"

_biomedicines, 2024, doi:10.3390/biomedicines12071606_

Round 1

Reviewer 1 Report

Comments and Suggestions for Authors

I compliment the authors on the highly detailed description of cited papers.

However, I suggest a more remarkable synthesis in Discussion and Conclusions to help the reader get a final picture from the many already reported, sometimes conflicting, results.

The language should be checked here and there for misspellings and/or potential misinterpretation by the reader. This is especially true for lines 60-65 and 161-175, which should be made more reader-friendly.

Please check the appropriate utilization of the FPG acronym thoroughly before para 2.5: sometimes, FPG erroneously substitutes for IFG, which can be confusing.

In lines 446-447, please expand a little on the rough features of HOMA-IR, considering different laboratory standards and the proposed initial 2.77 cut-off for normal subjects.

Regarding the sentence reported in lines 286-294, please comment on the study's possible intrinsic, insufficient statistical power due to the low number of patients within each subgroup.

Finally, what do you mean by "a non-systematic review which was preferred in order to achieve a more flexible approach and cover more data"? Please explain better.

Comments on the Quality of English Language

The language requires extensive revision here and there

Author Response

Response to Review 1 Comments

Dear Reviewer,

Thank you very much for your time and your effort to review our manuscript.

We are very grateful for providing your valuable feedback on the article.

Here is our response and related amendment that has been made in the manuscript according to your review (marked in yellow color).

I compliment the authors on the highly detailed description of cited papers.

Thank you very much. We really appreciate it!

However, I suggest a more remarkable synthesis in Discussion and Conclusions to help the reader get a final picture from the many already reported, sometimes conflicting, results.

Thank you very much.

We extended the Discussion section; the synthesis has been done as a take-home message as following:

“4.1. Most data come from retrospective (n = 14) and cross-sectional studies (n =13) and only three prospective studies were identified.

4.2. The analyzed population showed that the female population was slightly more frequent (the female to male ratio of 1.47), but the results in the glucose profile did not show a specific gender-related burden in these adults.

4.3. MACS prevalence amid NFAIs was 10 to 30%. Most studies sustain a higher T2DM prevalence in MACS (12 to 44%) versus NFAIs.

4.4. However, a few studies also showed a similar rate in NFAIs (up to 45%) and MACS or a higher T2DM rate than seen in healthy controls.

4.5. Prediabetes (such as IFG or IGT) may be more frequent in MACS versus NFAIs or NFAIs versus controls (no homogenous results).

4.6. Four studies introduced a CS sub-group and, paradoxically, only half of them confirmed a more severe glucose profile versus MACS/NFAIs.

4.7. The longest period of follow-up with concern to the glycaemic profile was of 10.5 years, and one cohort showed a significant increase in the T2DM rate at 17.9% compared to the baseline value of 0.03%.

4.8. Inconsistent data coming from six studies enrolling 1,039 individuals that underwent adrenalectomy (N = 674), respectively, conservative management (N = 365) pinpointed the impact of adrenalectomy in NFAIs that improved the regulation of the glucose metabolism after adrenalectomy versus baseline, respectively, versus conservative management (n = 3).

4.9. Noting these data, awareness of the T2DM and other glucose profile anomalies in NFAIs/MACS represents the key factor as transversal and longitudinal approach.”

Thank you very much

The language should be checked here and there for misspellings and/or potential misinterpretation by the reader. This is especially true for lines 60-65 and 161-175, which should be made more reader-friendly.

Thank you very much. We corrected them.

Please check the appropriate utilization of the FPG acronym thoroughly before para 2.5: sometimes, FPG erroneously substitutes for IFG, which can be confusing.

Thank you very much. We corrected them.

In lines 446-447, please expand a little on the rough features of HOMA-IR, considering different laboratory standards and the proposed initial 2.77 cut-off for normal subjects.

Thank you very much. We added this information

Regarding the sentence reported in lines 286-294, please comment on the study's possible intrinsic, insufficient statistical power due to the low number of patients within each subgroup.

Thank you very much. We added this very useful observation.

Finally, what do you mean by "a non-systematic review which was preferred in order to achieve a more flexible approach and cover more data"? Please explain better.

Thank you very much. We added a larger explanation.

Comments on the Quality of English Language: The language requires extensive revision here and there.

Thank you very much. We corrected them.

Thank you very much.

Reviewer 2 Report

Comments and Suggestions for Authors

The authors have analyzed the reported data on adrenal incidentalomas, cortisol secretion, diabetes, and glucose metabolism. The rationale is adequate, although I was expecting either a global analysis of the different reports or a meta-analysis in which the metabolic characteristics would have risk would be analyzed by clinically valuable parameters. On the other hand, the analysis performed is longer and more complicated to interpret, but it is suitable. The gender risk was informed in the abstract, but not the age risk, which is a critical factor in the cohort analyzed. In addition, it is essential to define if plasma glucose may be used or not as a parameter due to the problems encountered with Glycosilated Hb. The discussion could be improved with ftuture perspectives in the field. The conclusions are adequate.

Author Response

Response to Review 2 Comments

Dear Reviewer,

Thank you very much for your time and your effort to review our manuscript.

We are very grateful for your insightful comments and observations, also, for providing your valuable feedback on the article.

Here is a point-by-point response and related amendments that have been made in the manuscript according to your review (marked in yellow color).

The authors have analyzed the reported data on adrenal incidentalomas, cortisol secretion, diabetes, and glucose metabolism. The rationale is adequate, although I was expecting either a global analysis of the different reports or a meta-analysis in which the metabolic characteristics would have risk would be analyzed by clinically valuable parameters.

Thank you very much. We mentioned this aspect as limitation of the current work at the end of Discussion section: “Of note, this represents a very useful mathematical model of highlighting insulin resistance in daily practice, being considered the most common method of assessing insulin sensitivity which is easy-to-use and it only requires a fasting blood sample; notably, different lab standards and cut-offs varied over the time in normal subjects.” Thank you.

On the other hand, the analysis performed is longer and more complicated to interpret, but it is suitable. The gender risk was informed in the abstract, but not the age risk, which is a critical factor in the cohort analyzed.

Thank you very much. The age ranges were specified within the Abstract and Results section “aged between 14 and 96 years”. All tables included data with concern to the age within the studied population such as the ranges, mean, SD (depending on the study). The age-based analysis is very interesting; yet, we found no specific sub-analysis in this specific matter according to our methods in the mentioned studied other than the data we already presented. Thank you

In addition, it is essential to define if plasma glucose may be used or not as a parameter due to the problems encountered with Glycosilated Hb.

Thank you very much. We did not identify any study to address this very interesting issue of comparing the practical applicability of FPG as single input in addition or not with glycated hemoglobin, and specified this aspect at Discussion. Thank you

The discussion could be improved with future perspectives in the field.

We extended this section. Thank you

The conclusions are adequate.
